# MoSA: Motion-Coherent Human Video Generation via Structure-Appearance Decoupling

**Haoyu Wang**[1,3], **Hao Tang**[1], **Donglin Di**[3], **Zhilu Zhang**[4], **Wangmeng Zuo**[4],
**Feng Gao**[2], **Siwei Ma**[1], **Shiliang Zhang**[1]*

[1] State Key Laboratory of Multimedia Information Processing, School of Computer Science,
Peking University, Beijing, China
[2] School of Arts, Peking University, Beijing, China
[3] Li Auto, Beijing, China,
[4] Harbin Institute of Technology, Harbin, China
`haoyu.wang@stu.pku.edu.cn, haotang@pku.edu.cn,`
`donglin.ddl@gmail.com, cszlzhang@outlook.com, wmzuo@hit.edu.cn,`
`gaof@pku.edu.cn, swma@pku.edu.cn, slzhang.jdl@pku.edu.cn`

## Abstract

Existing video generation models predominantly emphasize appearance fidelity while exhibiting limited ability to synthesize complex human motions, such as whole-body movements, long-range dynamics, and fine-grained human–environment interactions. This often leads to unrealistic or physically implausible movements with inadequate structural coherence. To conquer these challenges, we propose MoSA, which decouples the process of human video generation into two components, i.e., structure generation and appearance generation. MoSA first employs a 3D structure transformer to generate a human motion sequence from the text prompt. The remaining video appearance is then synthesized under the guidance of this structural sequence. We achieve fine-grained control over the sparse human structures by introducing Human-Aware Dynamic Control modules with a dense tracking constraint during training. The modeling of human–environment interactions is improved through the proposed contact constraint. Those two components work comprehensively to ensure the structural and appearance fidelity across the generated videos. This paper also contributes a large-scale human video dataset, which features more complex and diverse motions than existing human video datasets. We conduct comprehensive comparisons between MoSA and a variety of approaches, including general video generation models, human video generation models, and human animation models. Experiments demonstrate that MoSA substantially outperforms existing approaches across the majority of evaluation metrics.

## 1 Introduction

General human video generation from text or image prompts (Jiang et al., 2023; Song et al., 2024; Wang et al., 2025a; Zhang et al., 2024b; Huang et al., 2024a) has recently garnered substantial research interest due to its broad application potentials. A core challenge lies in maintaining the structural plausibility of the human body, particularly for complex motions such as whole-body dynamics, long-range movement, and human–environment interactions, while preserving appearance fidelity in the generated videos. (Chefer et al., 2025).

Existing video generation models (Kong et al., 2024; Yang et al., 2024; Wan et al., 2025) often lack explicit guidance from human structural priors and are typically trained with noise reconstruction objectives in pixel space. Previous studies (Chefer et al., 2025; Jeong et al., 2024) have demonstrated that this paradigm leads to an overemphasis on appearance fidelity while neglecting human structural coherence, resulting in unrealistic human motion in the generated videos, as shown in

---

*Corresponding author.

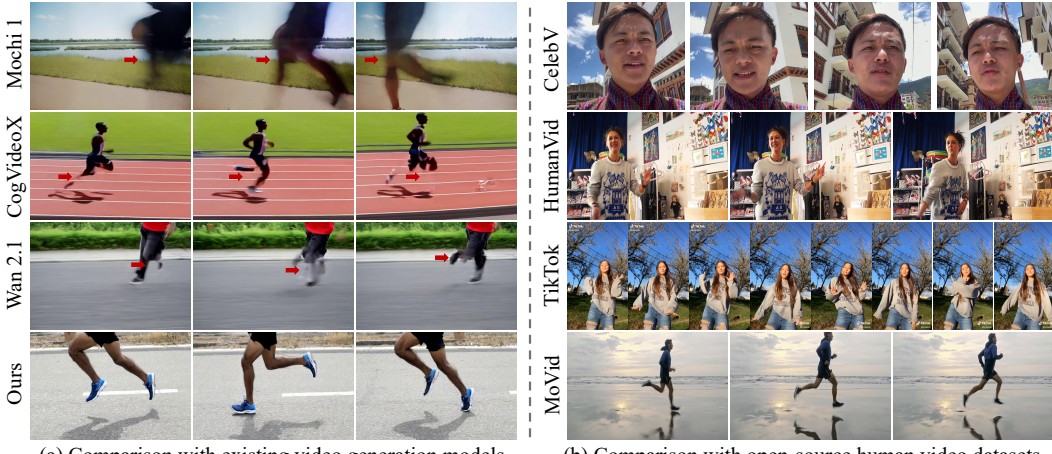

(a) Comparison with existing video generation models      (b) Comparison with open-source human video datasets

Figure 1: Illustration of the motivation. (a) shows sampled frames from videos generated with the prompt "running", where existing works (Team.; Yang et al., 2024) struggle to generate human videos with reasonable structures. (b) compares existing human video datasets (Wang et al., 2024d; Jafarian & Park, 2021) and our Movid, where existing datasets mostly focus on facial or upper-body regions, or consist of vertically oriented dance videos. More samples Movid are provided in Fig. 7 and supplementary materials.

Fig. 1(a). Since human appearance and motion convey different cues, they should adhere to different generation paradigms. This intuition leads to our MoSA, which generates human video through decoupling the structure and appearance generation. Specifically, as it is difficult to generate human videos with complex motions directly from the given text prompt, we first generate human motion structures conditioned on the text prompt. Given the sparse motion structures, MoSA subsequently synthesizes the visual appearance.

To generate the human motion structures, we first generate 3D human keypoints via a 3D structure transformer, which is pretrained on large-scale human motion datasets (Guo et al., 2022; Zhang et al., 2025b; Plappert et al., 2016). 3D human keypoint sequences are hence projected into a 2D skeleton sequence. Compared with directly producing 2D structural representations (Huang et al., 2024a; Song et al., 2024; Wang et al., 2025a), such as skeleton sequences, leveraging 3D human keypoints presents better robustness and accuracy because: i) 3D structure transformer leverages human priors to efficiently generate human keypoints, thereby ensuring the plausibility of the predicted human structure, and ii) by operating in 3D space, it can exploit implicit depth information to maintain structural plausibility in the presence of limb occlusions.

The subsequent video appearance is then synthesized under the guidance of this structural sequence. As the skeleton representation inherently provides only sparse structural guidance, its capability for fine-grained supervision in subsequent appearance generation is limited. To address issue, we propose the Human-Aware Dynamic Control module. It employs learnable dynamic weight predictors to generate weight maps corresponding to the skeleton features, hence further refines these maps using a tailored mask loss. This mask loss encourages the propagation of sparse skeleton guidance across the entire motion region and assigns dynamic weights to different spatial locations, thereby enhancing the fine-grained controllability of the sparse skeleton. In addition, previous studies (Chefer et al., 2025; Jeong et al., 2024) have shown that relying solely on the noise prediction objective during training may cause models to favor appearance fidelity over motion coherence. To mitigate this issue, we introduce a dense tracking loss aimed at enhancing the model's ability to preserve coherent motion structures. We further incorporate a contact constraint to accurately model human–environment interactions.

Besides the above methodology, this paper also contributes a novel human video dataset presenting complex human motions. Most existing human video datasets (Yu et al., 2023; Wang et al., 2024d; Li et al., 2024) primarily capture facial and upper-body movements with relatively simple movements, as illustrated in Fig. 1(b). Similarly, existing dance datasets (Jafarian & Park, 2021) exhibit

restricted background diversity and motion complexity, which confines corresponding generation approaches (Hu, 2024; Hu et al., 2025; Zhu et al., 2024; Gan et al., 2025; Zhang et al., 2024b) to dance videos and often requires auxiliary pose inputs. The fact that most existing open-source human video datasets (Wang et al., 2024d; Li et al., 2024; Jafarian & Park, 2021; Yu et al., 2023) primarily focus on simple movements, makes models trained on such datasets struggle to generate realistic and physically plausible motions. We thus introduce MoVid, a novel dataset comprising 30K human motion videos exhibiting diverse action categories and complex motions.

We employ MoVid as the training set for MoSA, and conduct comprehensive comparisons between MoSA and a broad spectrum of baselines, including general video generation models, human video generation models, and human animation models. The results demonstrate that MoSA substantially outperforms existing methods across most evaluation metrics, achieving superior performance in measures such as FVD, CLIP similarity, and VBench scores. As shown in Fig. 1(a), our method presents more reasonable body structure and more fluent motion.

In summary, our key contributions lies in three aspects: i) This is an original effort on structure–appearance decoupling framework for human video generation. As shown in extensive experiments, disentangling structural consistency from appearance synthesis benefits physically plausible human video generation. ii) Our proposed modules like Human-Aware Dynamic Control, dense tracking loss, and contact constraint lead to an effective implement of the proposed decoupling framework. They work well to enhance the fine-grained structural guidance, the modeling of motion coherence, as well as human–environment interactions. iii) A large-scale dataset MoVid is constructed to offer more diverse and complex motions than existing datasets. Extensive experiments demonstrate the superior performance of our method. The code and dataset will be released.

## 2 RELATED WORK

**Human Video Generation.** Most existing human video generation approaches rely on additional inputs beyond the text prompt, such as reference images of the target person (He et al., 2024; Yuan et al., 2024; Zhang et al., 2025a;c; Cao et al., 2025), driving pose sequences (Hu, 2024; Liu et al., 2025; Hu et al., 2025; Wang et al., 2024d; Gan et al., 2025; Zhang et al., 2024b; Zhu et al., 2024), or speech conditions (Sun et al., 2025; Lin et al., 2025; Tian et al., 2025; Meng et al., 2024; Cui et al., 2024). Among these works, ID-Animator (He et al., 2024) introduced a framework capable of generating identity-specific videos from reference facial images, along with a corresponding identity-oriented dataset. Building upon this foundation, subsequent studies have further advanced the task. For instance, ConsisID (Yuan et al., 2024) proposed a frequency decomposition strategy to improve identity fidelity. Moreover, AnimateAnyone (Hu, 2024) introduces an additional skeleton sequence as input to animate static human images. Building on this, AnimateAnyone2 (Hu et al., 2025) extends the framework to support environment-aware generation, enabling more coherent background migration. AnimateAnywhere (Liu et al., 2025) incorporates a camera motion learner to model background movement, thereby enhancing the realism of generated videos. However, the aforementioned methods are typically constrained to generating minor facial or upper-body movements, or are specialized for vertically oriented dance videos.

Some recent work (Song et al., 2024; Wang et al., 2025a; Huang et al., 2024a; Liang et al., 2025) has focused on more general text-driven human motion video generation, but due to the limitations of human video datasets (Wang et al., 2024d; Li et al., 2024), it is also difficult to generate realistic and physically compliant motion. To address these challenges, we propose a structure–appearance decoupling framework for generating motion-coherent human videos, and construct a large-scale dataset MoVid, to support the learning of complex human motion.

**Human Motion Generation.** Text-driven human motion generation aims to produce a sequence of human keypoints conditioned on a given text prompt, which can then be transformed into structural representations such as skeletons. Several existing approaches (Zhang et al., 2023a; Chen et al., 2023; Tevet et al., 2022; Zhang et al., 2024a; Yuan et al., 2023; Meng et al., 2025; Guo et al., 2024; Fan et al., 2025) leverage models trained on 3D motion-annotated datasets (Guo et al., 2022; Plappert et al., 2016; Lin et al., 2023; Zhang et al., 2025b) to generate 3D keypoint sequences. For example, MLD (Chen et al., 2023) extends latent diffusion models to support text-to-motion generation, while T2M-GPT (Zhang et al., 2023a) employs a generative pre-trained transformer (Radford et al., 2018) as the backbone and utilizes a vector-quantized variational autoencoder (Van Den Oord et al.,

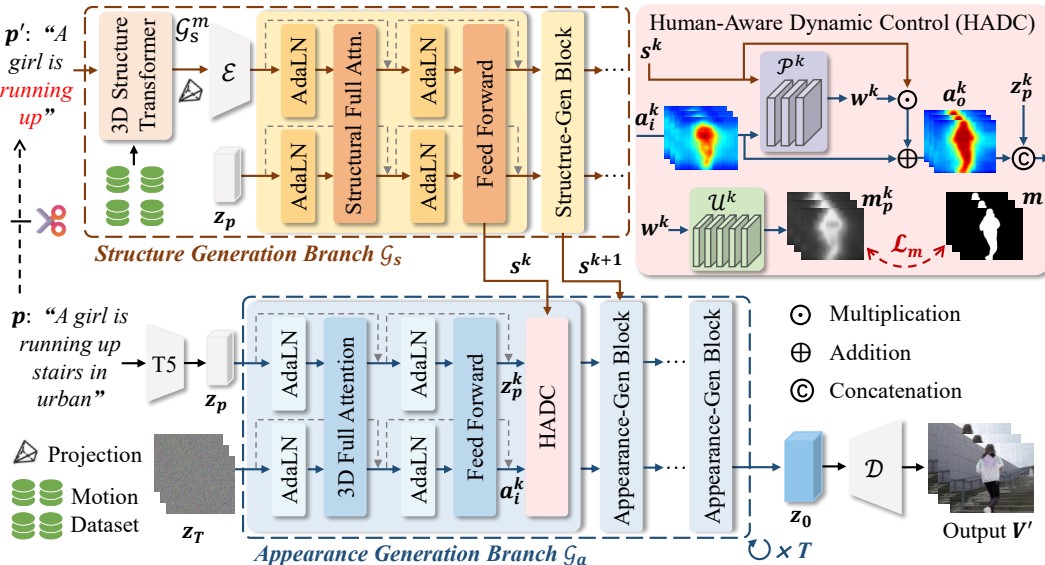

Figure 2: Overview of the proposed MoSA. Given a text prompt $p$, we first employ a 3D structure transformer to generate a structure sequence, which is subsequently encoded as structural features to guide the appearance generation. To further enhance motion consistency, we introduce human-aware dynamic control modules. For brevity, the Gate modules in blocks have been omitted.

2017) to encode and reconstruct keypoint features. In addition, several studies (Wang et al., 2025a; 2024c; Song et al., 2024) have explored directly generating 2D keypoints or skeleton sequences as representations of human motion.

These generated motion representations serve as structural priors for video generation and contribute to improve the plausibility of human motion. However, existing methods typically produce relatively sparse motion representations, limiting their capacity for fine-grained control. To address this, we introduce Human-Aware Dynamic Control modules that adaptively emphasize human-relevant regions and incorporate a dense tracking loss to further enhance the model's ability to learn structurally coherent and temporally coherent motion patterns.

# 3 METHODOLOGY

While recent video generation models (Yang et al., 2024; Kong et al., 2024; Wan et al., 2025) have achieved impressive visual quality, they frequently fail to generate physically plausible and structurally coherent human motion, especially in scenarios involving complex movement. Our MoSA aims to enhance structural consistency while preserving high-quality visual appearance, We present our method using the text-to-video setting as the primary example, while providing details of the image-to-video variant in Sec. H of the appendix.

Specifically, we begin by introducing the preliminaries of video generation models in Sec. 3.1. Afterwards, we describe the structure-appearance decoupling in detail in Sec. 3.2. To enhance the fine-grained controllability of sparse structural guidance, we propose the Human-Aware Dynamic Control (HADC) modules in Sec. 3.3. Finally, the training objectives are summarized in Sec. 3.4, which encompass the proposed dense tracking loss and contact constraint.

## 3.1 PRELIMINARY

**Video Generation Model.** Diffusion transformer (DiT) (Peebles & Xie, 2023) based generative models have attracted increasing attention due to their strong performance and scalability. Building on this (Yang et al., 2024; Wan et al., 2025), we adopt DiT as the backbone and extend it to support motion-coherent human video generation.

Given the text prompt $p$ and the corresponding video $V$, we first employ a pretrained T5 encoder (Raffel et al., 2020) to obtain the text embeddings $z_p \in \mathbb{R}^{B \times L \times D}$, where $B, L, D$ denote the batch size, token length and token dimensions, respectively. The video $V$ is encoded using a VAE encoder $\mathcal{E}$, and Gaussian noise $\epsilon$ is added to the resulting latent to obtain the noisy latent $z_v \in \mathbb{R}^{B \times F \times C \times H \times W}$, where $F, C, H, W$ denote the temporal, channel and spatial dimensions, respectively. The embeddings $z_p$ and latent $z_v$ are then fed into the backbone $\mathcal{G}_\theta$. The training objective is to learn a noise predictor $\mathcal{G}_\theta$ that estimates the added noise $\epsilon$. The loss function is formally defined as

$$\mathcal{L}_d = \mathbf{E}_{\epsilon, z_v, z_p, t} \left\| \epsilon - \mathcal{G}_\theta(\sqrt{\bar{\alpha}_t} z_v + \sqrt{1 - \bar{\alpha}_t}\epsilon, z_p, t) \right\|_2^2, \tag{1}$$

where $t$ is a random time step and $\alpha$ represents the predefined variance schedule.

During inference, a random Gaussian noise $z_T \sim \mathcal{N}(0, I)$ is sampled as $z_v$, and then $z_T$ is iteratively denoised through the backward process conditioned on the text embeddings $z_p$. The resulting latent $z_0$ is subsequently decoded by the VAE decoder $\mathcal{D}$ to generate the output video $V'$.

## 3.2 STRUCTURE-APPEARANCE DECOUPLING

Our method decouples the generation process into two branches, i.e., structure generation and appearance generation as illustrated in Fig. 2. Following parts proceed to provide a detailed description of this decoupled generation paradigm.

### 3.2.1 STRUCTURE GENERATION BRANCH

Given a text prompt $p$, the structure generation branch $\mathcal{G}_s$ aims to produce a human motion structure that aligns with the motion semantics conveyed by $p$. Since the prompt $p$ may also include appearance-related descriptions such as details of the surrounding environment, which are irrelevant to motion structure, we preprocess the input by extracting a motion-specific subset of $p$, denoted as $p'$. As illustrated in Fig. 1, $p'$ retains only motion-relevant information. This filtering process can be performed automatically using a Large Language Model (Yang et al., 2025) or specified by the user. The resulting motion-specific prompt $p'$ is then used as the input to the structure generation branch.

This branch is dedicated to generating human structures from the motion-specific prompt $p'$ without incorporating appearance information. However, directly training a text-driven model to produce 2D structural sequences, such as skeletons, often fails to guarantee the anatomical plausibility of the generated structures. We thus reformulate the text-driven structure generation task as a 3D keypoint sequence generation task. This formulation could i) leverage human priors to efficiently generate $K$ human keypoints, thereby ensuring the plausibility and coherence of the predicted human structure, and ii) benefit from the generation in 3D space. In other words, it can utilize implicit depth information to preserve structural consistency in scenarios involving limb occlusions. Once the keypoint sequence is obtained, it is rendered into 2D space. Following common practice in conditional human video generation (Hu et al., 2025; Gan et al., 2025), we convert the keypoint sequence into a skeleton representation, which serves as the final structural guidance $g_s$. The above process can be formally described as follows:

$$g_s = \text{Projection}(\mathcal{G}_s^m(z_T^s, p')), \tag{2}$$

where $\mathcal{G}_s^m$ denotes the 3D structure transformer, and $z_T^s$ represents Gaussian noise sampled from a standard normal distribution $\mathcal{N}(0, I)$. Following previous work (Fan et al., 2025; Meng et al., 2025), $\mathcal{G}_s^m$ adopts the same autoregressive architecture and is first pretrained on million-scale motion datasets. Details are described in Sec. A of the appendix.

After obtaining $g_s$, we employ it as an additional control signal encoding human structural information to guide subsequent appearance generation. To effectively encode and incorporate this condition into the appearance generation process, we introduce specialized structure generation blocks within the DiT architecture. The overall process can be formalized as follows:

$$s^{1:N} = \mathcal{G}_s(\mathcal{E}(g_s)), \tag{3}$$

where $s^k$ $(k = 1, ..., N)$ represents the output of the $k$-th structure generation block in $\mathcal{G}_s$, which is used to guide the subsequent appearance generation, and $N$ is the total number of such blocks.

### 3.2.2 Appearance Generation Branch

The appearance generation branch $\mathcal{G}_a$ is designed to synthesize realistic video content conditioned on $p$ and $s^{1:N}$, capturing both the environmental appearance and human subjects, while preserving the realism of human motion. As described in Sec. 3.1, a pretrained T5 encoder is employed to extract the text embedding $z_p$ from the prompt $p$, and the initial latent $z_T$ is sampled from $\mathcal{N}(0, I)$, both of which are served as the input of this appearance generation branch.

To enhance the controllability of structure guidance, particularly in the context of sparse skeleton representations, we introduce the Human-Aware Dynamic Control (HADC) modules within this branch. After $T$ steps of iterative denoising, the resulting latent $z_0$ is decoded by a VAE decoder $\mathcal{D}$ (Kingma, 2013), yielding human videos $V'$ with coherent motion and high-fidelity appearance that align with the semantics of the text prompt $p$.

### 3.3 Human-Aware Dynamic Control

The human structure features $s^{1:N}$ can be utilized as auxiliary conditions in $\mathcal{G}_a$ to enhance the plausibility of human motion. However, the structural guidance $g_s$, typically represented as a sparse skeleton, lacks the expressiveness required for fine-grained motion control.

To address this limitation, Human-Aware Dynamic Control (HADC) modules, which are inserted between adjacent DiT blocks within the appearance generation branch. Each $k$-th HADC module takes the structural signal $s^k$, the intermediate video latent $a_i^k$, and the text embedding $z_p^k$ produced by the preceding DiT block as input. By leveraging the structural cues embedded in $s^k$, the HADC module refines $a_i^k$ to enable fine-grained control over human motion, producing motion-enhanced latents $a_o^k$ that are subsequently passed to the next DiT block. Specifically, we design a human-aware dynamic weights predictor $\mathcal{P}^k$, which aims to i) facilitate the propagation of $s^k$ throughout the whole motion region in $a_i^k$ and ii) assign spatially-varying control weights to these human motion regions within $a_i^k$, i.e.,

$$w^k = \mathcal{P}^k(s^k, a_i^k), \tag{4}$$

where $w^k$ denotes the human-aware dynamic control weights. Leveraging $w^k$, the sparse skeleton feature $s^k$ can exert fine-grained control over the video latents, i.e.,

$$a_o^k = a_i^k \oplus (w^k \odot s^k), \tag{5}$$

where $a_o^k$ denotes the motion-enhanced video latents, while $\oplus$ and $\odot$ indicate element-wise addition and multiplication, respectively. In addition, to ensure the effectiveness of $w^k$, we design a learnable network $\mathcal{U}^k$ to convert $w^k$ into mask latents and constrain it through a mask loss $\mathcal{L}_m$ during training, i.e.,

$$\mathcal{L}_m = \sum_{k=1}^{N} \left\| \mathcal{U}^k(w^k) - \mathcal{E}(M) \right\|_2^2, \tag{6}$$

where $\mathcal{U}^k(w^k)$ denotes the predicted mask latent $m_p^k$, $M$ denotes the ground truth video mask, and $\mathcal{E}(M)$ is the corresponding latent $m$. By incorporating the proposed HADC modules, the consistency of human motion within the video latent $a_i^k$ is significantly improved, as illustrated in Fig. 2.

### 3.4 Training Objectives

During the training of these two branches, the pretrained 3D structure transformer $\mathcal{G}_s^m$ is excluded, and the skeleton sequence extracted from the ground truth video $V$ is directly used as the structural condition $g_s$. To further improve the model's capacity for learning temporally coherent motion, we introduce a dense tracking loss $\mathcal{L}_{track}$, i.e.,

$$\mathcal{L}_{track} = \frac{1}{\sum_{(t_v, t_v')}^{\mathcal{S}} e^{\frac{|t_v - t_v'|}{2}}} \sum_{(t_v, t_v')}^{\mathcal{S}} e^{\frac{|t_v - t_v'|}{2}} \cdot \left\| U'_{t_v \to t_v'} - U_{t_v \to t_v'} \right\|_1, \tag{7}$$

where $t_v, t_v'$ are video timestamps, $U'$ and $U$ represent the 2D tracks corresponding to the generated video $V'$ and the ground truth video $V$, respectively. These tracks are extracted by Co-Tracker3 (Karaev et al., 2024). $e^{\frac{|t_v - t_v'|}{2}}$ denotes a temporal weighting function that assigns higher

Table 1: Quantitative comparison with existing methods. Lower FVD values indicate better performance, whereas higher values on the other metrics correspond to better results. **Bold** indicates the best performance, and underline denotes the second-best.

| Method | FVD | CLIPSIM | Subject Consistency | Background Consistency | Motion Smoothness | Dynamic Degree | Imaging Quality |
|--------|-----|---------|--------------------|------------------------|-------------------|----------------|-----------------|
| ModelScope | 1945 | 0.2739 | 90.87% | 93.41% | 96.22% | 48.57% | 60.12% |
| VideoCrafter2 | 1959 | 0.2801 | 93.43% | 97.01% | 97.31% | 35.71% | 60.32% |
| LaVie | 1778 | 0.2895 | 93.80% | 95.51% | 97.21% | **53.73%** | 62.57% |
| Mochi 1 | 1207 | 0.2903 | 94.67% | 95.32% | 97.75% | 51.14% | 54.65% |
| CogVideoX | 1360 | 0.2899 | 93.75% | 94.02% | 97.78% | 51.42% | 62.98% |
| HunyuanVideo | 1235 | 0.2948 | 94.41% | 95.17% | 98.95% | 50.42% | 58.13% |
| Wan 2.1 | 1251 | 0.2951 | 94.43% | 95.55% | 98.36% | 51.71% | 65.21% |
| Ours | **1093** | **0.3035** | **96.83%** | **97.43%** | **99.25%** | 52.86% | **65.43%** |

Figure 3: Visual comparison with existing video generation models. For clarity, VideoCrafter2 (Chen et al., 2024) is denoted as VC2, and HunyuanVideo (Kong et al., 2024) is denoted as Hunyuan. It demonstrates that our method is capable of generating realistic human motion with with reasonable structures, while simultaneously preserving high fidelity in video appearance.

loss weights to longer time intervals, thereby encouraging the model to better capture long-range motion dependencies. $\mathcal{S}$ represents the set of time intervals, and $(t_v, t'_v) \in \mathcal{S}$, *i.e.*,

$$\mathcal{S} = \{(t_v, t'_v) \mid 0 \leq t_v < T_v, 0 \leq t'_v < T_v, t_v \neq t'_v\}, \tag{8}$$

Table 2: Effect on the Wan2.1 base model.

| Models | FVD | CLIPSIM |
|---|---|---|
| Wan 2.1 | 1251 | 0.2951 |
| + Our Decoupling Framework | **1108** | **0.3044** |

Table 3: Effect of the contact constraint.

| Constraint $\mathcal{L}_{cont}$ | FVD | CLIPSIM |
|---|---|---|
| ✘ | 1108 | 0.3021 |
| Ours | **1093** | **0.3035** |

Table 4: Effect of the decoupling framework.

| Structure Generation Branch | FVD | CLIPSIM |
|---|---|---|
| ✘ | 1262 | 0.2971 |
| 2D Structure Generation | 1230 | 0.2998 |
| Ours | **1093** | **0.3035** |

Table 5: Effect of the HADC modules.

| HADC Modules | FVD | CLIPSIM |
|---|---|---|
| ✘ | 1188 | 0.2973 |
| w/o $\mathcal{L}_m$ | 1112 | 0.3009 |
| Ours | **1093** | **0.3035** |

Table 6: Effect of the dense tracking loss $\mathcal{L}_{track}$.

| Dense Tracking Loss $\mathcal{L}_{track}$ | FVD | CLIPSIM |
|---|---|---|
| ✘ | 1172 | 0.3009 |
| Static Weights | 1114 | 0.3016 |
| Ours | **1093** | **0.3035** |

Table 7: Necessity of our MoVid.

| Training Dataset | FVD | CLIPSIM |
|---|---|---|
| ✘ | 1360 | 0.2899 |
| HumanVid | 1217 | 0.2949 |
| MoVid (Ours) | **1093** | **0.3035** |

where $T_v$ is the video length. Additionally, we propose a 3D contact constraint $\mathcal{L}_{cont}$ to further enhance the modeling of human–environment interactions. Due to page limitations, full details are presented in Sec. B of the appendix. The overall training objective is formulated as:

$$\mathcal{L} = \mathcal{L}_d + \lambda_m \mathcal{L}_m + \lambda_{track} \mathcal{L}_{track} + \lambda_{cont} \mathcal{L}_{cont}, \tag{9}$$

where $\lambda_m$, $\lambda_{track}$ and $\lambda_{cont}$ are the weights used to balance different loss terms.

## 4 EXPERIMENTS

### 4.1 EXPERIMENTAL DETAILS

**Datasets.** Existing human video datasets are largely restricted to simple movements, limiting models' ability to synthesize realistic and physically plausible motion in complex scenarios. To overcome these limitations, we curated MoVid, a dataset of 30K real-world human motion videos with annotations. Details are provided in Sec. C of the appendix. For evaluation, we collected more than 300 text prompts spanning motion types and environmental contexts following previous work.

**Evaluation Metrics.** Following previous work, we adopt Fréchet Video Distance (FVD) (Unterthiner et al., 2019) and CLIP similarity (CLIPSIM) (Radford et al., 2021) to measure the performance. Furthermore, we utilize VBench (Huang et al., 2024b) to conduct a more comprehensive assessment of model performance across multiple dimensions, including subject consistency, background consistency, motion smoothness, motion dynamics, and visual quality.

### 4.2 COMPARISON WITH EXISTING METHODS

We adopt CogVideoX-5B-T2V (Yang et al., 2024) as the backbone of the appearance generation branch, with additional implementation details provided in Sec. D of the appendix. We first compare our method with video generation models, including ModelScope (Wang et al., 2023), VideoCrafter2 (Chen et al., 2024), LaVie (Wang et al., 2024a), Mochi 1 (Team.), CogVideoX-5B-T2V (CogVideoX) (Yang et al., 2024), HunyuanVideo (Kong et al., 2024), and Wan2.1-T2V-14B (Wan 2.1) (Wan et al., 2025), accompanied by a user study (Sec. E.3). For text-driven human video generation methods (Huang et al., 2024a; Wang et al., 2025a; Song et al., 2024) without public implementations, we conduct qualitative comparisons based on their released videos, if available (Sec. E.2). We further compare MoSA with pose-driven human animation methods (Hu, 2024; Wang et al., 2024d; Men et al., 2025; Tu et al., 2025), with additional results and visualizations presented in Sec. E.5 of the appendix. More video types are shown in Sec. G.

*"A man dressed in athletic attire is stretching his legs in a park, silhouetted against a tree and a bridge."*

*"An elderly man with short, graying hair is performing exercise on a black mat in a well-equipped gym."*

Figure 4: Effect of our decoupling framework MoSA when applied to Wan 2.1 (Wan et al., 2025).

*"A man with long black hair, is seated on a blue rug in a modern living room, playing a classical guitar."*

*"A woman in a red sports bra and pink shorts performs a leg exercise in a gym, with one leg lifted high."*

Figure 5: Effect of structure-appearance decoupling. For experiments that employ the structure generation branch, we also visualize the corresponding generated human structure.

**Quantitative Comparison.** Quantitative comparisons with existing methods are shown in Tab. 1. Compared with previous models, our MoSA achieves excellent performance in various metrics.

**Qualitative Comparison.** Visual comparisons are presented in Fig. 3. As illustrated, our approach is capable of generating realistic human motion, both for basic actions such as walking and jumping, as well as for more complex activities like skating. In contrast, existing methods often struggle to generate physically plausible motion with coherent structural integrity for complex movements. More visual results are provided in Sec. E.1.

## 5 ABLATION STUDY

### 5.1 EFFECT OF OUR DECOUPLING FRAMEWORK WHEN APPLIED TO WAN 2.1

The proposed decoupled generation framework exhibits high compatibility with state-of-the-art video generation models and can be seamlessly integrated to enhance their performance. Specifically, the structure branch focuses on producing plausible and coherent motion, while the appearance branch leverages the intrinsic strengths of these models to synthesize realistic textures and environmental details. To further validate its effectiveness, we apply our MoSA to Wan 2.1 by incorporating the proposed components and finetuning the base model. As shown in Tab. 2 and Fig. 4, the results demonstrate both the transferability and the effectiveness of our MoSA framework.

### 5.2 EFFECT OF STRUCTURE-APPEARANCE DECOUPLING

Tab. 4 presents the analysis for the effect of structure-appearance decoupling. In the first row, the structure generation branch is removed, and the base model is directly finetuned using MoVid. The second row utilizes outputs from an independently trained 2D skeleton sequence generation model as the structure guidance $g_s$. Details of this model are described in Sec. F.1 of the appendix. Visual comparisons are provided in Fig. 5. Compared with 2D structure generation, our $\mathcal{G}_s^m$ effectively

preserves structure correctness, as demonstrated by the missing leg in the second-row on the left side of Fig. 5. Additionally, our $\mathcal{G}_s^m$ leverages the depth information in 3D space to maintain spatial coherence in scenarios involving limb occlusion. For example, in the second row on the right of Fig. 5, the right leg is incorrectly placed behind the left, leading to an implausible body structure.

### 5.3 EFFECTS OF OTHER PROPOSED COMPONENTS

We further investigate the effects of other proposed components, including the HADC modules (Tab. 5 and Sec. F.2), the dense tracking loss (Tab. 6 and Sec. F.3), the contact constraint (Tab. 3 and Sec. F.4), and the necessity of the MoVid dataset (Tab. 7 and Sec. F.5). Due to page limitations, detailed analyses and extended visual comparisons are presented in the Sec. F of the appendix.

## 6 CONCLUSION

We present MoSA, a structure–appearance decoupling framework for realistic human video generation. A 3D structure transformer synthesizes motion structures to guide appearance generation, while human-aware dynamic control and a dense tracking loss enhance fine-grained motion coherence. To better capture human–environment interactions, we introduce a contact constraint. Furthermore, we curate a large-scale human video dataset to overcome the limitations of existing datasets. Extensive experiments show that MoSA consistently outperforms previous approaches.

## 7 DISCUSSIONS AND FUTURE WORK

Generating complex inter-person contact and fine-grained hand interactions has long been a challenging problem for video generation models. Built upon a structure-appearance disentanglement paradigm, our method substantially improves the quality of multi-person interactions and fine-grained motion synthesis. As shown in Fig. 17, the girl's bowstring-pulling action in the second row and the two-person jumping and ball-contesting scene in the third row demonstrate clear enhancements.

Nevertheless, we found that generating highly intricate hand motions remains a challenge, where current methods may still produce artifacts such as distorted or blurred finger structures. However, our structure-appearance disentanglement paradigm is naturally compatible with incorporating denser structural cues, such as hand keypoints. This inherent compatibility offers a promising pathway to enhance the realism of finger-level actions by integrating such detailed guidance.

Our further investigation reveals that the primary obstacle lies in the training data. Existing motion datasets used for training 3D structure transformers typically include only SMPL body joints. Consequently, incorporating hand keypoints would necessitate augmenting these datasets with additional 3D annotations for hand joints to enable effective model training. We believe this is a very worthwhile direction to explore and will pursue it in our future work, with extending the proposed MoVid dataset.

## ETHICS STATEMENT

This work focuses on human video generation and the collection of a human motion dataset. All data were obtained from public sources. The dataset will be released for research under a license that prohibits misuse such as surveillance, deepfake creation, or other harmful applications. This research adheres to institutional ethical standards and legal regulations.

## REPRODUCIBILITY STATEMENT

We provide the core codes and data samples in the supplementary materials. The appendix provides additional implementation details of our work. Furthermore, the pre-processing steps for the datasets are described in the supplementary materials. The core codes and MoVid dataset will be released to the public upon final preparation.

ACKNOWLEDGMENTS

This work was supported in part by under Grant 2023-JCJQ-LA-001-088, in part by the Natural Science Foundation of China under Grant U20B2052, Grant 61936011, in part by under 2025ZD1601300, in part by the Okawa Foundation Research Award, in part by the Ant Group Research Fund, and in part by the Kunpeng&Ascend Center of Excellence, Peking University.

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

APPENDIX

The content of this appendix involves:

- Details of the 3D Structure Transformer in Sec. A.
- Details of the proposed 3D contact constraint in Sec. B, including its differential process.
- Description of the proposed MoVid dataset, including data cleaning, annotation, and statistic comparison in Sec. C.
- Implementation details of MoSA in Sec. D.
- More comparative experiments in Sec. E, including more comparisons with general video diffusion models, text-driven human video generation models, commercial video generation models, human animation models, and generation results with occlusion situations.
- More ablation studies in Sec. F, including details of the 2D structure generation model, final visualization of the HADC module outputs, effect of human-aware dynamic control modules, effect of dense tracking loss, effect of the contact constraint, necessity of the MoVid dataset, effect of the selected camera poses, and the effect of the data from different views in our dataset..
- More video types generated by MoSA in Sec. G.
- Image-to-video generation variant of MoSA in Sec. H.
- Evaluation on more motion-centric metrics in Sec. I.
- LLM usage declaration in Sec. J.

## A    DETAILS OF 3D STRUCTURE TRANSFORMER

We employ a 3D structure transformer $\mathcal{G}_s^m$ to generate 3D human keypoints. Following previous work (Fan et al., 2025; Meng et al., 2025), $\mathcal{G}_s^m$ adopts a unified autoregressive architecture and is pretrained on million-scale motion datasets (Guo et al., 2022; Fan et al., 2025), as illustrated in Fig. 6. These large-scale datasets encompass a wide variety of motion categories, including diverse and complex human movements, which enable robust pretraining. During inference,

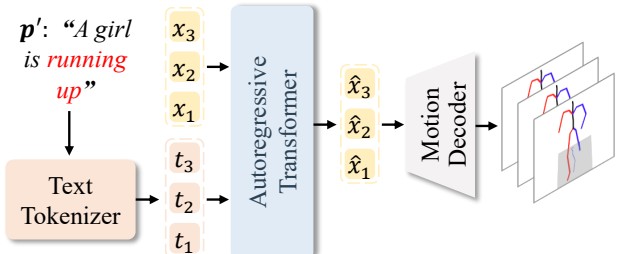

Figure 6: Architecture of the 3D structure transformer $\mathcal{G}_s^m$. The final skeleton frames are obtained after projection.

the model takes as input a motion-specific text prompt $p'$ and initial motion latents $x_{1:N}$. The prompt $p'$ is first encoded by a text tokenizer (Raffel et al., 2020) to obtain conditional embeddings, which, together with $x_{1:N}$, are fed into an autoregressive transformer (Vaswani, 2017) to predict the final motion latents. These latents are then decoded by the motion decoder into a human keypoint sequence aligned with the semantics of $p'$. Finally, a standardized skeleton sequence $g_s$ is derived from the predicted keypoints.

## B    DETAILS OF THE PROPOSED 3D CONTACT CONSTRAINT

To ensure physically plausible human–environment interactions, we introduce a 3D contact loss that penalizes unrealistic interpenetrations. The computation proceeds in four steps. First, each video frame is lifted into a 3D point cloud representation using a pretrained VGGT (Wang et al., 2025c) model. Formally, for the $f$-th frame, we obtain a point set $P_f = p_i \in \mathbb{R}^3$. Second, human and scene points are separated based on 2D segmentation masks. Each 3D point $p_i$ is projected onto the image plane using the intrinsic matrix $K_f$ and extrinsic matrix $E_f$ predicted by the VGGT model. If the projected pixel $(u_i, v_i)$ lies within the segmented human region $M_f$, the point is assigned to the human set $H_f$; otherwise, it is assigned to the scene set $S_f$. Third, scene points from all frames

are aggregated to reconstruct a mesh $M_b$ via convex hull estimation. This mesh is then converted into a signed distance function (SDF), denoted $SDF(\cdot)$, where negative values indicate locations inside the scene surface and positive values denote outside regions. Finally, the 3D contact loss is defined. If human points penetrate the scene ($SDF(h) < \tau$), the loss penalizes the penetration depth; otherwise, it encourages human points to remain close to the scene surface by minimizing their distance to the mesh. The resulting loss is formally expressed as:

$$
\mathcal{L}_{cont}^{f} = \begin{cases} \sum_{h \in H_f^-} \left| SDF(h) - \tau \right|, & \text{if } |H_f^-| > 0, \\ \min_{h \in H_f} SDF(h) - \tau, & \text{otherwise,} \end{cases}
\tag{10}
$$

where $H_f^- = \{h \in H_f \mid SDF(h) < \tau\}$ denotes the set of penetrating points, and $\tau$ is the penetration threshold (set to zero by default). The final contact loss $\mathcal{L}_{cont}$ is calculated by summing the values across all frames and then normalizing by the number of frames. By incorporating the constraint $\mathcal{L}_{cont}$ during training, we effectively mitigate issues such as penetration and other physically implausible behaviors that frequently arise in the generation of complex human–environment interactions.

**Differential Process.** For the f-th frame, the 3D contact loss is defined as Eq. 10.

The propagation path is as follows, where $h$ denotes a human point, $V'$ denotes the generated video, $\theta$ denotes the model parameters.

$$
\mathcal{L}_{cont}^{f} \rightarrow \text{SDF}(h) \rightarrow h \rightarrow V' \rightarrow \theta
\tag{11}
$$

(1) The process begins with the loss function $\mathcal{L}_{cont}^{f}$ itself. Since $\tau$ is set to 0, for a human point $h$, the loss form can be simplified to:

$$
\mathcal{L}_{cont}^{f,h} = \max(0, -\text{SDF}(h))
\tag{12}
$$

(2) The partial derivative of this loss with respect to the SDF value is non-zero only in the case of interpenetration:

$$
\frac{\partial \mathcal{L}_{cont}^{f,h}}{\partial \text{SDF}(h)} = \begin{cases} -1, & \text{if } \text{SDF}(h) < 0, \\ 0, & \text{otherwise.} \end{cases}
\tag{13}
$$

(3) The next step is to propagate the gradient from the SDF value to the 3D coordinates $(x, y, z)$ of the point $h$. The gradient of a scalar field like the SDF with respect to a point's coordinates is a vector:

$$
\nabla_h \text{SDF} = \frac{\partial \text{SDF}(h)}{\partial h}
\tag{14}
$$

This gradient vector provides the most efficient direction to displace the point $h$ in order to move it towards the exterior of the scene mesh, thereby resolving the penetration.

Using the chain rule, we combine the gradients from the previous two steps to compute the gradient of the loss with respect to the 3D coordinates of the human point $h$:

$$
\frac{\partial \mathcal{L}_{cont}^{f,h}}{\partial h} = \frac{\partial \mathcal{L}_{cont}^{f,h}}{\partial \text{SDF}(h)} \cdot \frac{\partial \text{SDF}(h)}{\partial h}
\tag{15}
$$

(4) Then, the gradient $\frac{\partial \mathcal{L}_{cont}^{f,h}}{\partial h}$ can be backpropagated through the layers of VGGT [2] to obtain the gradient with respect to its input, i.e. the generated video frame $V'$:

$$
\frac{\partial \mathcal{L}_{cont}^{f,h}}{\partial V'} = \frac{\partial \mathcal{L}_{cont}^{f,h}}{\partial h} \cdot \frac{\partial h}{\partial V'}
\tag{16}
$$

(5) Finally, the generated video frame $V'$ is the output of our model, $V' = \text{MoSA}(\text{inputs}; \theta)$. With the gradient $\frac{\partial \mathcal{L}_{cont}^{f,h}}{\partial V'}$ computed, we can continue the backpropagation through the MoSA network to

obtain the gradients with respect to all of its learnable parameters $\theta$:

$$\frac{\partial \mathcal{L}_{cont}^{f,h}}{\partial \theta} = \frac{\partial \mathcal{L}_{cont}^{f,h}}{\partial V'} \cdot \frac{\partial V'}{\partial \theta} \tag{17}$$

These final gradients are then used by the optimizer to update the model's parameters.

## C IN-DEPTH DESCRIPTION OF THE MOVID DATASET

### C.1 DATA CLEANING AND ANNOTATION

Following HumanVid (Wang et al., 2024d), we construct the MoVid dataset by collecting a large number of source videos from public sources (Pexels.; YouTube.) using motion–related keywords. To ensure data quality, we employ the "Motion Smoothness" "Dynamic Degree" and "Image Quality" metrics from VBench (Huang et al., 2024b) to filter out low-quality samples. Coarse-grained textual annotations are then generated using CogVLM2 (Hong et al., 2024), followed by manual verification to ensure accuracy. For spatial supervision, we apply SAM (Kirillov et al., 2023) to obtain human masks and utilize DWPose (Yang et al., 2023) to extract human keypoints and skeletons. To ensure motion richness, we compute the average offset $\bar{o}$ of each keypoint across the entire video and discard samples with $\bar{o} \leq 0.1$. Additionally, videos containing only simple motions, such as isolated facial or upper-body movements, are excluded, retaining examples that exhibit complex dynamics. Through this rigorous pipeline, we curate a dataset comprising approximately 30K high-quality real-world video clips encompassing more diverse and complex human motions and scenes.

### C.2 DATASET STATISTICS

Tab. 8 provides a detailed comparison with several representative real-world human video datasets, including CelebV-HQ (Zhu et al., 2022), CelebV-Text (Yu et al., 2023), TikTok (Jafarian & Park, 2021), UBC-Fashion (Zablotskaia et al., 2019), IDEA-400 (Lin et al., 2023) and HumanVid (Wang et al., 2024d). Most existing datasets, such as CelebV (Yu et al., 2023; Zhu et al., 2022), Human-Vid (Wang et al., 2024d), and the recent OpenHumanVid (Li et al., 2024), predominantly focus on facial or upper-body motions. In addition, other datasets are often constrained by limited motion diversity or specific video formats. For example, TikTok (Jafarian & Park, 2021) and UBC-Fashion (Zablotskaia et al., 2019) primarily consist of vertically oriented videos with restricted motion ranges. Others, such as IDEA-400 (Lin et al., 2023), lack fine-grained and accurate textual annotations, which limits their applicability in text-conditioned generation tasks. To address these limitations, we introduce the MoVid, a high-quality whole-body human video dataset with millions of frames. Examples are provided in **dataset_sample.zip** of the provided supplementary materials, and are also shown in Fig. 7.

Tab. 8 also shows the comparison in terms of action complexity and action types. For Action Complexity, inspired by VMBench, we first segment the regions related to human and calculate the average optical flow of pixels in these regions. We utilize SEA-RAFT (Wang et al., 2024b) to estimate optical flow, which serves as a measure of the magnitude of human motion within the dataset. Specifically, smaller optical flow values correspond to subtler movements, whereas larger values are indicative of more significant and intricate motion. Results demonstrate that our proposed MoVid dataset has more complex human actions. For Action Types, CelebV-HQ and CelebV-Text are restricted to facial movements, while the TikTok dataset focus on a singular category of dance. The UBC-Fashion dataset, in turn, is composed exclusively of standing persons exhibiting only subtle motion. For the IDEA-400, HumanVid, and our proposed MoVid datasets, we first generate text annotations for videos using CogVLM2, and then analyze these annotations to identify all verb and verb-object phrases related to human, calculating the number of unique action types in each dataset. The results show that our MoVid dataset has a richer variety of human motion. The above results provide strong evidence that the MoVid dataset encompasses a broader, more diverse, and more complex spectrum of human motion videos.

Table 8: Comparison of MoVid with existing representative real-world human video datasets.

| Dataset | Clips | Resolution | Action Types ↑ | Action Complexity ↑ | Fine-grained Caption |
|---|---|---|---|---|---|
| CelebV-HQ | 35K | 512×512 | Facial type | 0.6891 | - |
| CelebV-Text | 70K | 512×512 | Facial type | 0.7070 | Text |
| TikTok | 340 | 604×1080 | Dancing | 0.6816 | - |
| UBC-Fashion | 500 | 720×964 | Standing | 0.3321 | - |
| IDEA-400 | 12K | 720P | 5K | 0.5969 | - |
| HumanVid | 20K | 1080P | 7K | 0.6669 | - |
| MoVid (Ours) | 30K | 1080P | 17K | 1.1124 | Text |

Video samples of our MoVid Dataset

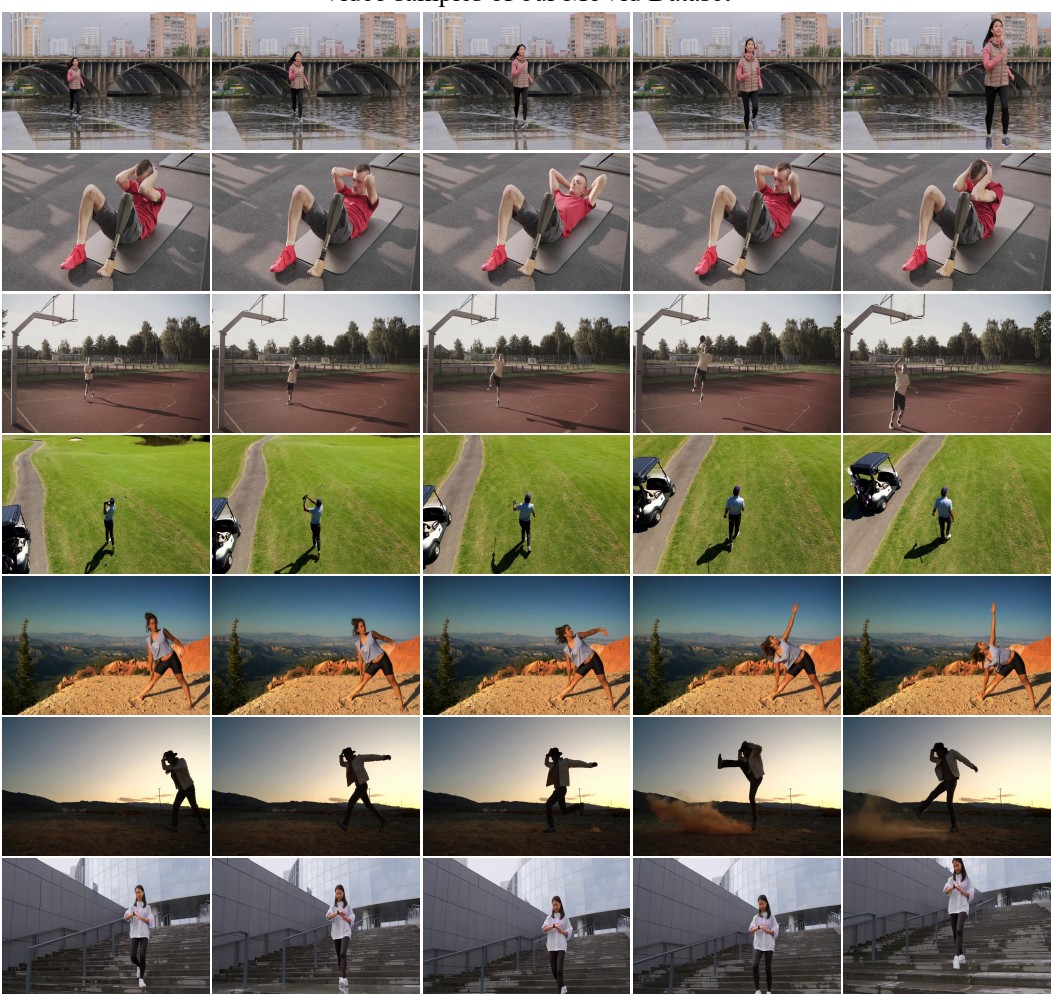

Figure 7: Examples randomly selected from the proposed MoVid dataset.

# D  IMPLEMENTATION DETAILS OF MoSA

We use CogVideoX-5B-T2V (Yang et al., 2024) as the base model of the appearance generation branch. During training, we freeze the original weights and train only the structure generation blocks and HADC modules. We also present the results for the version using Wan 2.1 (Wan et al., 2025) as the base model in Tab. 2 and Fig. 4, following the same training settings as described above. For the proposed HADC modules, $\mathcal{P}^k$ comprises three linear layers interleaved with two activation

*"A person is repeatedly lifting dumbbells in a well-lit gym filled with mirrors and fitness equipment ..."*

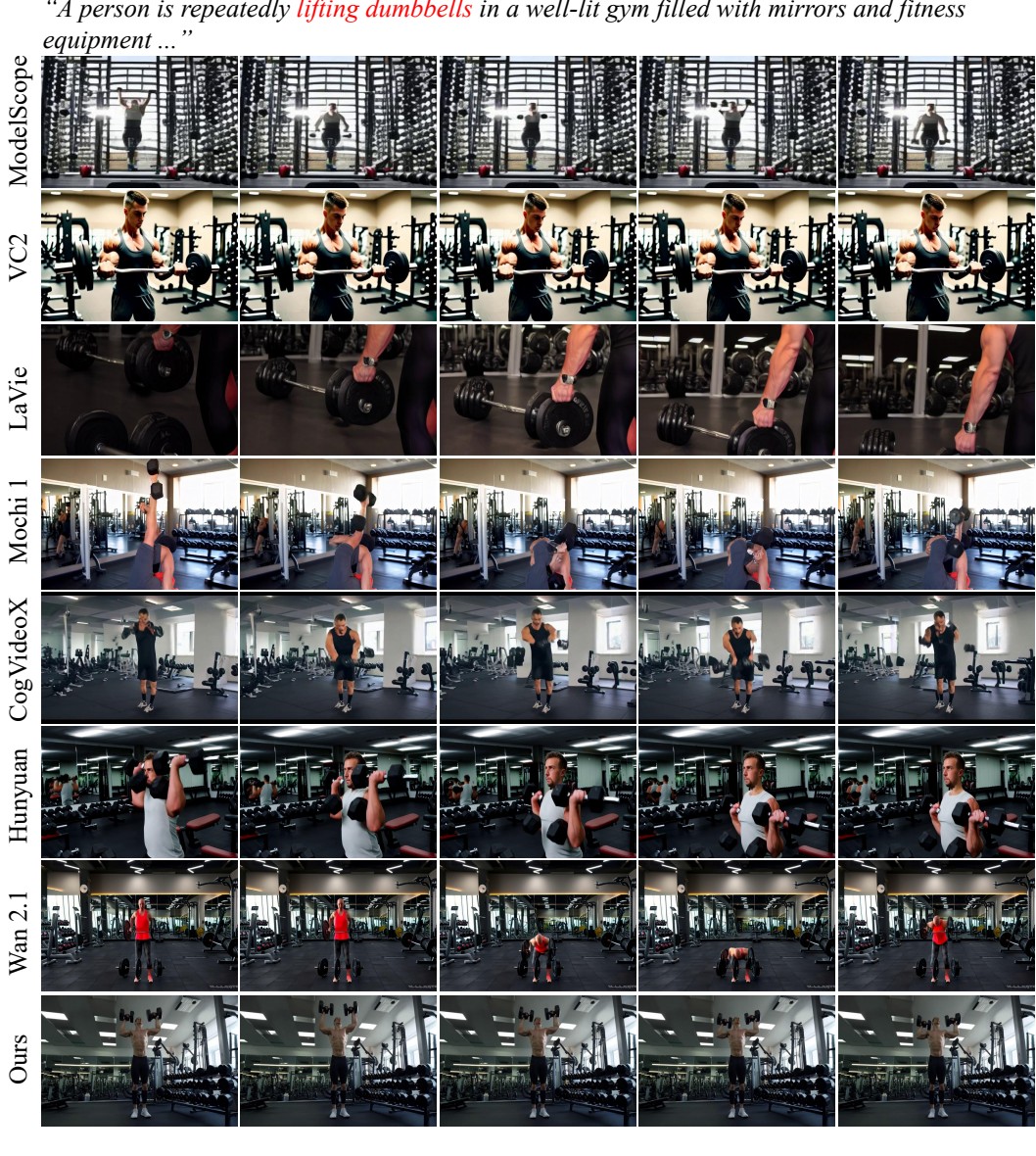

Figure 8: More visual comparison with existing video generation models.

functions (Hendrycks & Gimpel, 2016; Wang et al., 2025b; Zhang et al., 2023b). $\mathcal{U}^k$ shares a similar architecture with $\mathcal{P}^k$, but includes an additional up-sampling layer followed by a 3D convolution layer (Tran et al., 2015) at the end. During training, the skeleton sequence is extracted from the input video $V$. We train and test at a resolution of 720×480, and train for 20,000 iterations on 4 NVIDIA A800 GPUs with a batch size of 16. The AdamW (Loshchilov & Hutter, 2017) optimizer is used with a learning rate of 1e-5. We set the loss weights $\lambda_m$, $\lambda_{track}$ and $\lambda_{cont}$ to 0.001, 0.01 and 10.0, respectively.

## E MORE COMPARATIVE EXPERIMENTS

### E.1 VISUAL COMPARISON WITH EXISTING VIDEO GENERATION MODELS

We provide more visual comparison with ModelScope (Wang et al., 2023), VideoCrafter2 (Chen et al., 2024), LaVie (Wang et al., 2024a), Mochi 1 (Team.), CogvideoX (Yang et al., 2024), Hun-yuanVideo (Kong et al., 2024) and Wan 2.1 (Wan et al., 2025) in Fig. 8 and Fig. 9. We also show the

*"A young man stands on an indoor basketball court, holding a basketball with both hands at his chest ..."*

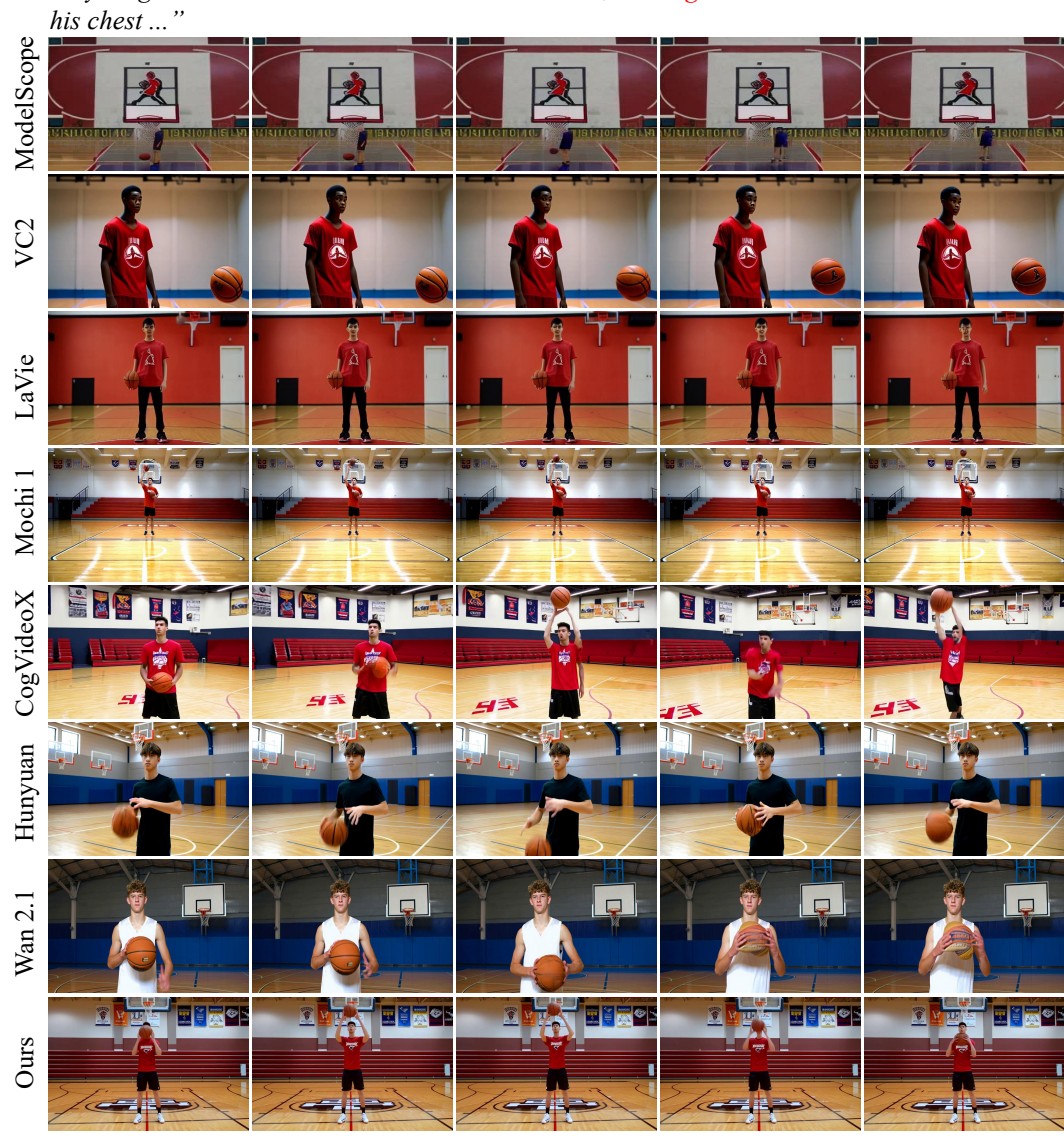

Figure 9: More visual comparison with existing video generation models.

video results in the provided video.mp4. The **video.mp4** in the supplementary materials presents additional and more compelling visual comparisons. As illustrated, MoSA demonstrates superior capability in generating human videos with coherent and physically plausible motion compared to existing approaches. Additionally, we alson showcase a broader range of generation results to highlight the diversity and generalization ability of MoSA in video.mp4.

### E.2 VISUAL COMPARISON WITH TEXT-DRIVEN HUMAN VIDEO GENERATION METHODS

Since existing text-driven human video generation methods (Wang et al., 2025a; Huang et al., 2024a; Song et al., 2024) are not yet open source, we perform a visual comparison based on their released videos, if accessible. Fig. 11 presents a visual comparison with Move-in-2D (Huang et al., 2024a) and HumanDreamer (Wang et al., 2025a). As shown, our method generates more realistic and visually coherent results, demonstrating improved fidelity and motion consistency. The video comparison with Move-in-2D and HumanDreamer is shown in video.mp4.

*"An elderly man with short, graying hair is performing exercise on a black mat in a well-equipped gym ..."*

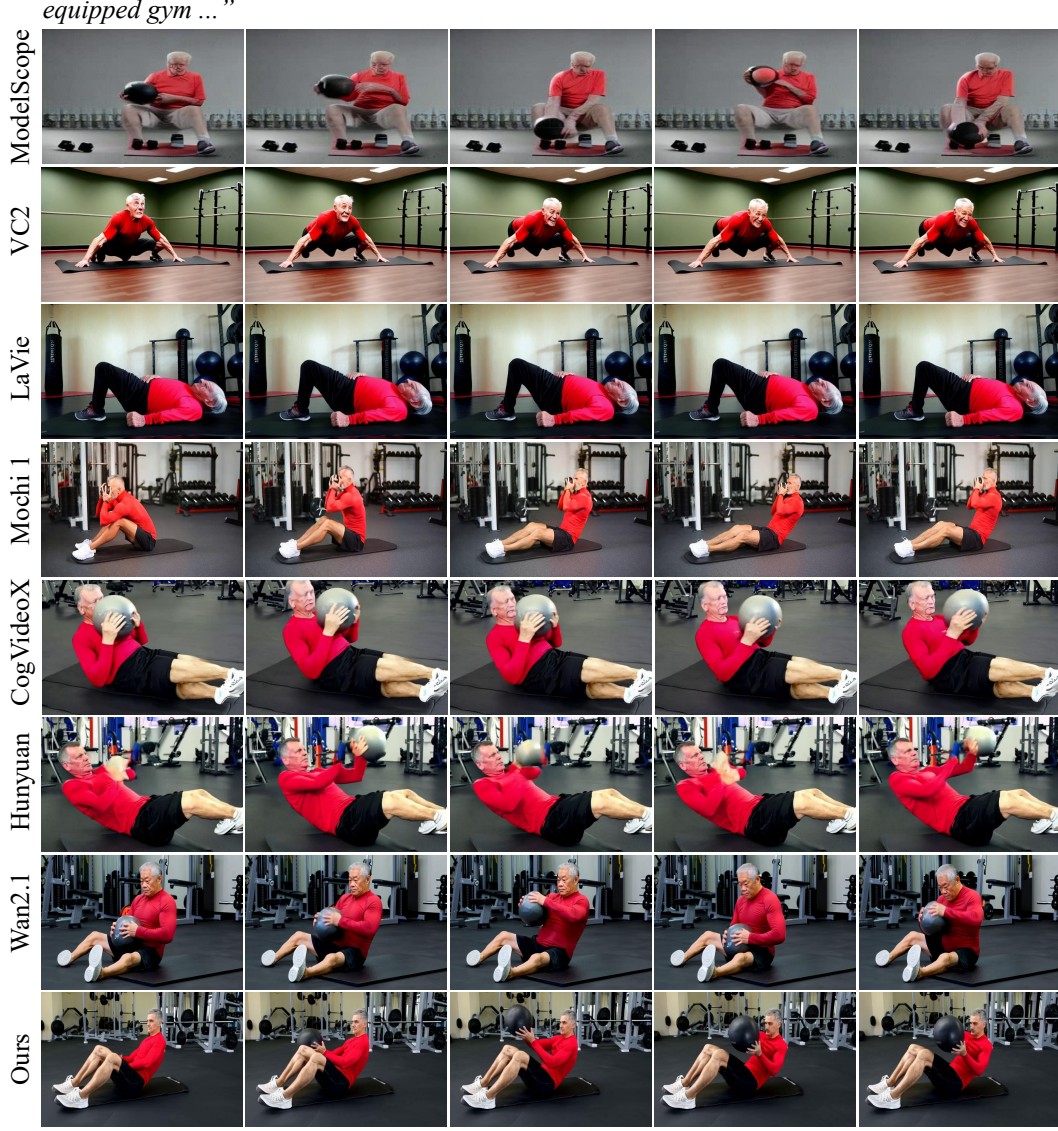

Figure 10: More visual comparison with existing video generation models.

### E.3    USER STUDY

To enable a more comprehensive evaluation, we conduct a manual assessment of the generated results across different methods. Specifically, participants are presented with video samples generated by various models for each text prompt and are asked to select the most preferred video based on two criteria: Motion Quality, which assesses the realism and coherence of human motion, and Video Quality, which evaluates the overall visual fidelity and realism of the generated appearance. We then calculated the proportion of times each method is selected as the best. As reported in Tab. 9 and Tab. 10, our MoSA achieved the highest preference rates in both Motion Quality and Video Quality, demonstrating its superior performance in generating visually realistic and motion-coherent human videos.

*"A man hitting a tennis return."*

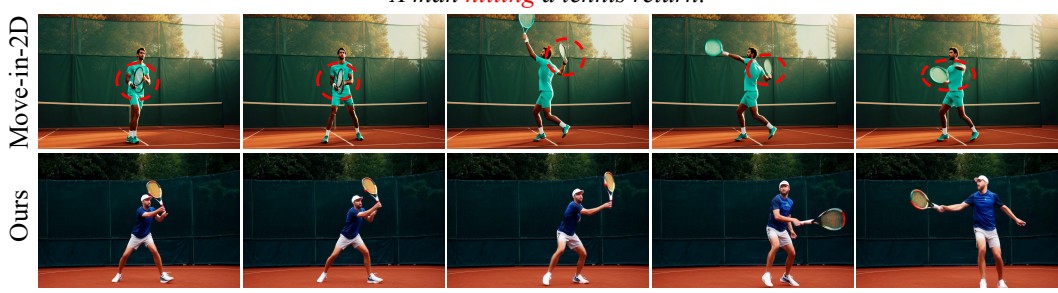

*"A man is performing a yoga pose on a mat, and he is seen moving his legs ..."*

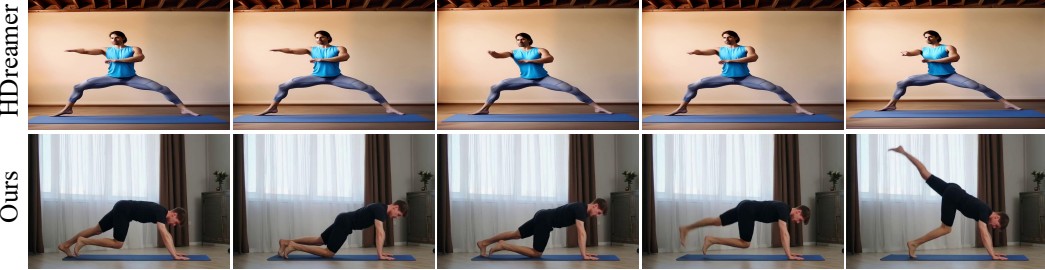

Figure 11: Visual comparison with text-driven human video generation models Move-in-2D and HumanDreamer (HDreamer) on their released samples.

Table 9: User study with existing video generation models. Motion Quality assesses the realism and plausibility of human motion, while Video Quality evaluates the overall perceptual quality of the generated videos.

| Method | Motion Quality ↑ | Video Quality ↑ |
|---|---|---|
| ModelScope (Wang et al., 2023) | 3.91% | 1.92% |
| VC2 (Chen et al., 2024) | 4.14% | 2.33% |
| LaVie (Wang et al., 2024a) | 10.62% | 5.75% |
| Mochi 1 (Wang et al., 2024a) | 8.38% | 10.34% |
| CogVideoX (Yang et al., 2024) | 12.07% | 16.01% |
| Hunyuan (Kong et al., 2024) | 15.21% | 15.55% |
| Wan 2.1 (Wan et al., 2025) | 15.41% | 18.98% |
| Ours | **30.26%** | **29.12%** |

### E.4 COMPARISON WITH THE COMMERCIAL MODELS OF KLING AND SEEDANCE

To better illustrate the performance of our MoSA model, we provide a comparison with Kling and Seedance in Fig. 12. Given that Kling and Seedance are commercial models requiring a queue for access, we are only able to evaluate their performance within a limited timeframe using text prompts from our paper. We observe that despite the impressive capabilities of commercial models like Kling and Seedance, they are still susceptible to the issue of human anatomical distortion.

### E.5 COMPARISON WITH HUMAN ANIMATION METHODS

We test the performance of human animation methods, including Animate Anyone (Hu, 2024), HumanVid (Wang et al., 2024d), MIMO (Men et al., 2025) and StableAnimator (Tu et al., 2025), as shown in Tab. 11. These methods use the same 2D skeletons in MoSA as input, with reference images generated from the text prompts. The results further demonstrate the superiority of our method. Furthermore, we train an additional I2V version of our MoSA on the human animation task in which the 3D structure transformer is removed during both training and testing. Following Human-Vid (Wang et al., 2024d), we then compare this version with pose-driven human video generation

Table 10: User study with Move-in-2D and HumanDreamer on their released videos.

| Method | Motion Quality ↑ | Video Quality ↑ |
|---|---|---|
| Move-in-2D (Huang et al., 2024a) | 24.3% | 24.7% |
| HumanDreamer (Wang et al., 2025a) | 26.2% | 23.4% |
| Ours | **49.5%** | **51.9%** |

*"A man stretches on a wooden pier, placing one hand on his foot and the other reaching towards his leg."*

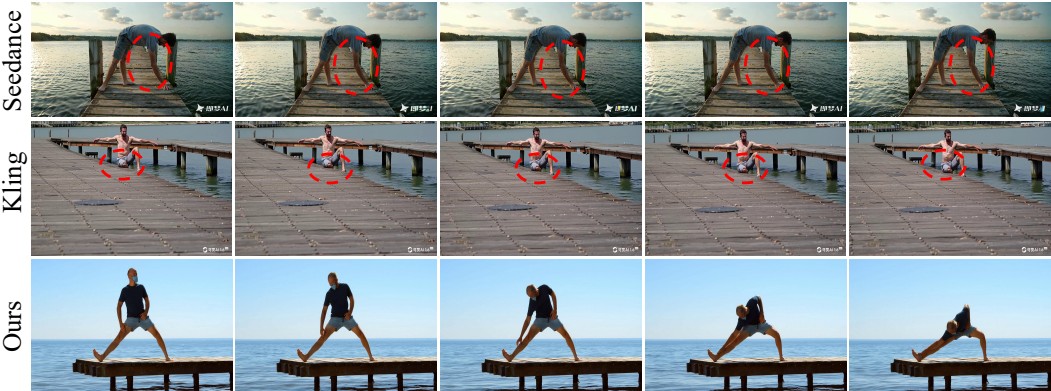

*"A young woman, clad in a sleeveless beige top, stretches her leg on a gravel path in a tranquil park."*

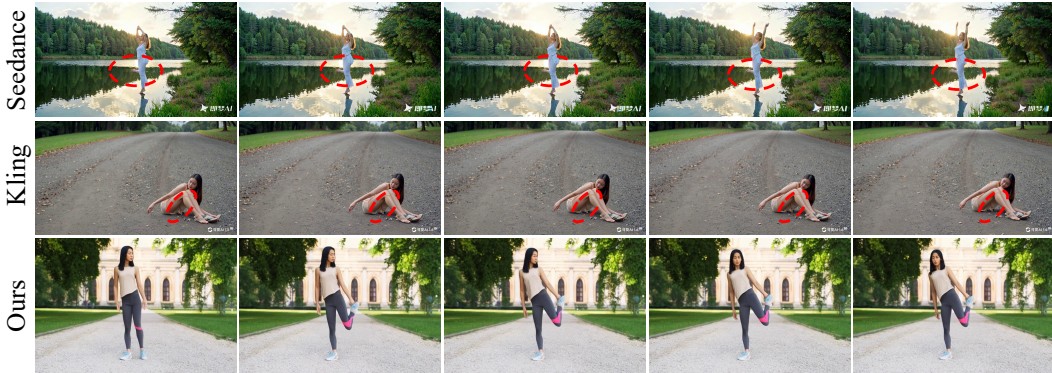

Figure 12: Visual comparisons with Kling and Seedance. The watermark in the bottom right corner is automatically added by their models.

methods using the same training and test sets on Humanvid (Wang et al., 2024d), as well as identical prompts and pose conditions. Results are presented in the Tab. 12, which further demonstrate the superiority of our method. Since MIMO and StableAnimator are trained on self-collected in-house datasets, they are not included in these tables.

### E.6 GENERATION RESULTS WITH OCCLUSION SITUATIONS

The core role of the HADC modules is to enhance the model's fine-grained control over human motion. During training, we introduce a mask loss $L_m$ calculated from the ground-truth human masks. This process guides the model to learn how to effectively propagate the sparse structural guidance $s$ to the corresponding **visible regions** of the human body in the image. Through this supervision, the model learns the association between the skeletal structure and the actual visible body parts.

Table 11: Comparison with existing human animation methods on our test sets for human video generation in general scenarios.

| Method | FVD | CLIPSIM | Subject Consistency | Background Consistency | Motion Smoothness | Dynamic Degree | Imaging Quality |
|---|---|---|---|---|---|---|---|
| Animate Anyone | 1362 | 0.2850 | 94.09% | 95.33% | 97.23% | 41.28% | 57.06% |
| HumanVid | 1374 | 0.2876 | 95.12% | 94.77% | 97.42% | 42.34% | 54.05% |
| MIMO | 1285 | 0.2904 | 94.82% | 95.49% | 97.38% | 45.57% | 53.83% |
| StableAnimator | 1326 | 0.2895 | 95.37% | 94.89% | 97.88% | 42.85% | 56.96% |
| Ours | **1108** | **0.3021** | **97.74%** | **97.37%** | **99.31%** | **52.86%** | **64.68%** |

Table 12: Comparison with pose-driven methods on the HumanVid dataset.

| Method | SSIM ↑ | PSNR ↑ | LPIPS ↓ | FVD ↓ | FID ↓ |
|---|---|---|---|---|---|
| Animate Anyone (Hu, 2024) | 0.602 | 16.108 | 0.368 | 1248.4 | 97.74 |
| Champ (Zhu et al., 2024) | 0.653 | 15.028 | 0.426 | 1985.2 | 100.59 |
| HumanVid (Wang et al., 2024d) | 0.672 | 19.534 | 0.275 | 732.7 | 46.06 |
| Uni3C (Cao et al., 2025) | 0.687 | 20.779 | 0.221 | **562.7** | 35.11 |
| Ours | **0.691** | **20.802** | **0.209** | 568.3 | **34.76** |

Consequently, this mechanism naturally handles scenarios with partial occlusions. During training, if a part of the human body is occluded by an object (e.g., a leg hidden behind a table), the corresponding ground-truth human mask for that area will be absent. The model learns that even when the structural guidance $s$ indicates that a leg should be present, it should avoid generating the leg's appearance in regions where it is actually occluded. In other words, the HADC module learns to render the visual appearance only in the **non-occluded areas** corresponding to the structural guidance. This capability is demonstrated on the left side of Fig. 5 in our paper, where our model properly handles a person being occluded by objects.

To further substantiate the effectiveness of our approach, we add additional visual results in Fig. 18 in the appendix. These results showcase more diverse occlusion scenarios and clearly demonstrate that MoSA can generate high-quality and physically plausible videos even when partial occlusions are present.

## F MORE ABLATION STUDIES

### F.1 DETAILS OF THE 2D STRUCTURE GENERATION

To assess the effectiveness of the 3D Human Structure Generator, we conducted an ablation study as reported in Tab. 4 of the main paper. The "2D Structure Generation" row indicates that the output of an additionally trained 2D skeleton generation model is used as $g_s$. Specifically, we fine-tune the base model CogVideoX-5B using text-skeleton pairs from the MoVid dataset. However, this approach introduces two major issues: (1) directly generating 2D skeletons often compromises structural plausibility, as exemplified by the result on the left side of Fig. 5 in main paper, where a leg is entirely missing; (2) 2D representations struggle to resolve ambiguities arising from limb occlusions, frequently leading to physically implausible poses, such as the example on the right side of Fig. 5, where the right leg is incorrectly generated behind the left. In contrast, the 3D Human Structure Generator effectively mitigates these issues by leveraging depth information and human priors.

### F.2 EFFECT OF HUMAN-AWARE DYNAMIC CONTROL

Tab. 5 presents the quantitative analysis of the HADC modules. The first row indicates discarding the entire HADC modules, and the second row indicates discarding only $\mathcal{L}_m$. We can see that they could improve the model's performance. To further illustrate their effect, the corresponding visual results are shown in Fig. 13(a). When HADC modules are removed, the generated human structure remains

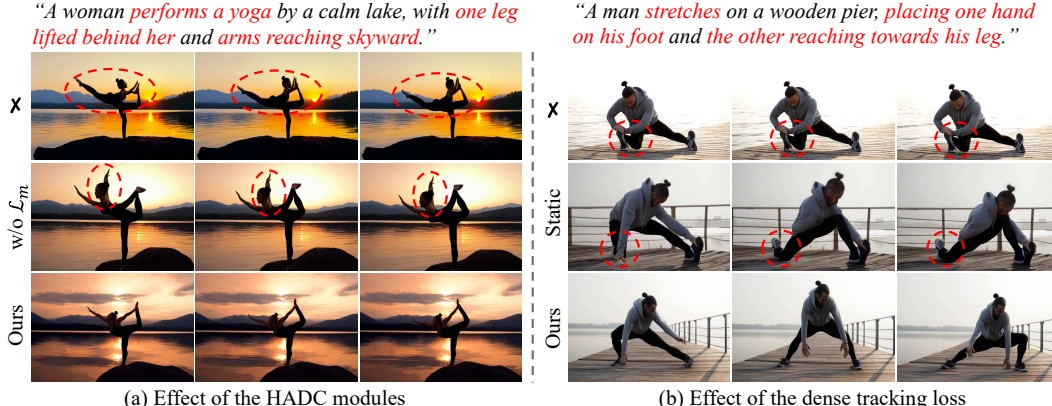

(a) Effect of the HADC modules    (b) Effect of the dense tracking loss

Figure 13: Effect of the HADC modules and dense tracking loss. "Static" means applying a fixed weight.

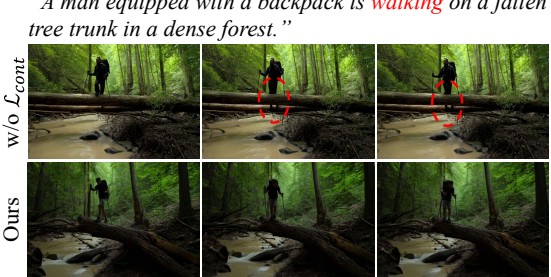

Figure 14: Visual effect of the proposed contact constraint $\mathcal{L}_{cont}$.

roughly aligned with the expected body layout but lacks fine-grained motion control. Without $\mathcal{L}_m$, details of corresponding human motion will also be unrealistic.

### F.3    EFFECT OF THE DENSE TRACKING LOSS

Tab. 6 and Fig. 13(b) present the effect of the dense tracking loss. "Static weights" means applying a fixed weight to different time intervals. By introducing temporal tracking optimization, the model's ability to learn temporally coherent motion is enhanced. Furthermore, assigning greater loss weights to motion pairs with longer temporal intervals further improves the overall consistency of human motion across time.

### F.4    EFFECT OF THE CONTACT CONSTRAINT

Tab. 3 and Fig. 14 present the effect of the proposed contact constraint. As illustrated in Fig. 14, the absence of the proposed contact constraint leads to noticeable interpenetration when the human walks on a fallen tree trunk. In contrast, our method, with the constraint incorporated, generates more realistic motions without penetration artifacts. This highlights the effectiveness of the contact constraint in modeling physically plausible human–environment interactions.

### F.5    NECESSITY OF THE MOVID DATASET

To explore the necessity of the proposed MoVid dataset, we conduct corresponding experiments, as shown in Tab. 7. The first row represents the performance of the base model without any additional human video dataset for training. The second row represents the performance after fine-tuning with the existing open-source human video dataset. We selected the widely used HumanVid dataset with various human motions and expanded it to the same scale as MoVid. Fig. 15 illustrates the visual effects. As most existing datasets primarily capture simple motion like facial expressions or upper-

Figure 15: Necessity of the MoVid dataset.

Table 13: Results of different camera poses when rendering the 3D human structure.

| Method | FVD | CLIPSIM |
|---|---|---|
| Fixed Camera 1 | 1093 | 0.3035 |
| Fixed Camera 2 | 1089 | 0.3032 |
| Time-varying Camera 1 | 1106 | 0.3029 |
| Time-varying Camera 2 | 1096 | 0.3031 |

body movements, they fall short in supporting the generation of more complex motions. In contrast, models trained on the MoVid dataset are able to synthesize physically plausible and structurally coherent human motions. This is due to the fact that the proposed MoVid dataset covers a wide variety of motion types and more complex dynamics, so that the model trained on it can capture a variety of motions in the real world, enhancing the model's ability to generate realistic and physically plausible human motion videos.

### F.6 EFFECT OF THE SELECTED CAMERA POSES WHEN RENDERING THE 3D STRUCTURE

During inference, users can freely specify camera poses, including elevation angle, azimuth angle, and distance from the center of the coordinate system. These camera poses can be either fixed or time-varying. The quantitative results reported in the paper are obtained by rendering the 3D human pose from a fixed camera view. To further evaluate the impact of camera poses, we provide additional quantitative results under two different fixed camera poses and two time-varying camera trajectories. It is important to note that this does not require retraining the model. The corresponding results are presented in Tab. 13, demonstrating that our method is robust to variations in the rendering views of the generated 3D human structures.

In addition, for the camera movements of the background, we build on the capabilities of the Cogvideo-X to generate coherent camera movements based on text descriptions and video content. Visual results are shown in Fig. 16.

### G MORE VIDEO TYPES GENERATED BY MOSA

In Fig. 17, we show examples of half-body and multi-person results generated by our MoSA. For half-body results, our method allows users to specify which keypoints to retain when rendering the 3D pose, thereby generating partial body skeletons. For example, by discarding lower body keypoints (knees, feet, etc.) during rendering, a skeleton containing only the upper body can be generated. For multi-person results, during inference, we can optionally use the 3D structure transformer to generate multi-person structures, based on which we generate subsequent videos.

Dynamic camera trajectory for human

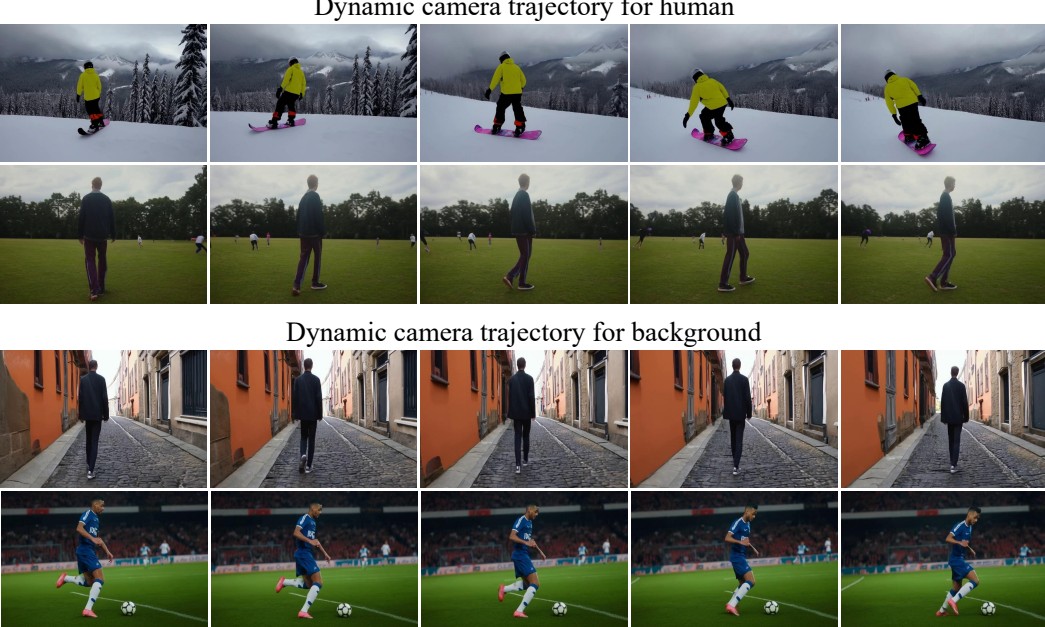

Dynamic camera trajectory for background

Figure 16: Visual results of dynamic camera trajectory for human and background.

## H  IMAGE-TO-VIDEO GENERATION VARIANT OF MOSA

We additionally train an image-to-video generation variant based on the CogVideoX-5B-I2V (Yang et al., 2024). During training, a frame is randomly sampled from the ground-truth video to serve as the image prompt, while the text prompt is retained as input. Moreover, data augmentation is applied to the skeleton conditions by randomly translating and scaling the input skeleton sequences, enhancing the model's generative capability. During inference, the 3D structure transformer $\mathcal{G}_s^m$ first produces the corresponding skeleton sequence based on the text prompt. The text prompt, image prompt, and skeleton sequence are then fed into corresponding branches to generate the target video. Visualization results are presented in Fig. 17.

## I  EVALUATION ON MORE MOTION-CENTRIC METRICS

We design more motion-centric metrics and add these evaluation metrics to the ablation study of the structure-appearance decoupling paradigm, as shown in Table 14. Furthermore, we use these metrics to compare existing T2V models with our approach, and the corresponding results are presented in Table 15. The Pose Confidence metric is computed by applying the DWPose [1] model to extract human keypoints from the generated videos and averaging their confidence scores. Higher confidence indicates more accurate pose estimation, which further reflects the plausibility of the generated human poses, avoiding structural failures such as missing limbs. Temporal Smoothness is derived by calculating the mean jerk of tracking points within human regions, where lower values correspond to smoother motion. The Contact-Violation metric is evaluated using Qwen3-VL, a powerful multimodal model equipped with extensive world knowledge. It identifies physically implausible artifacts such as interpenetrations in the generated videos and computes the corresponding violation rate. Action Recognizability is also assessed using Qwen3-VL, which determines whether the generated actions faithfully match those specified in the text prompt, such as the text prompt "walks to the blue trampoline and jumps" in Fig. 3. Motion Plausibility and Action Correctness are the evaluation criteria used in the user study.

Moreover, we have conducted a user study, as reported in Table 9 of the paper, to assess the motion quality of the generated videos. To more comprehensively demonstrate the advantages of our approach, we further evaluate the motion plausibility and action correctness of the generated videos,

Half-body videos and multi-person videos generated by our MoSA

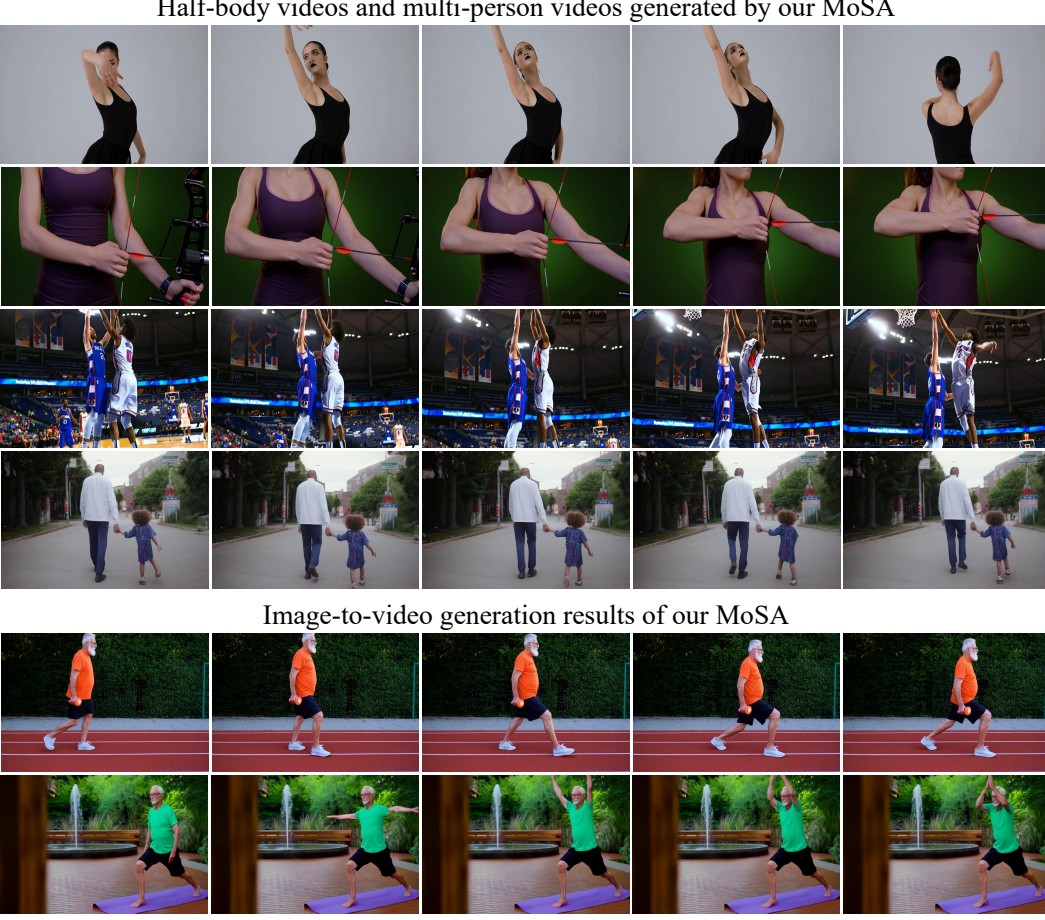

Image-to-video generation results of our MoSA

Figure 17: Half-body and multi-person results generated by our MoSA. We also provide image-to-video generation results in this figure.

with the corresponding results presented in Table 16. These motion-centric metrics provide additional evidence that our method exhibits clear superiority in generating high-quality human motion.

Table 14: Effect of the decoupling paradigm, evaluating on more motion-centric metrics.

| Structure Branch | Pose Confidence | Temporal Smoothness | Contact-Violation | Action Recognizability | Motion Plausibility | Action Correctness |
|---|---|---|---|---|---|---|
| ✗ | 0.885 | 0.122 | 2.04% | 96.85% | 5.09% | 8.24% |
| 2D Structure Generation | 0.896 | 0.114 | 1.60% | 98.42% | 15.13% | 15.60% |
| Ours | **0.911** | **0.103** | **1.08%** | **99.45%** | **79.78%** | **76.16%** |

## J  LLM USAGE DECLARATION

Large Language Models (ChatGPT) were used exclusively to improve the clarity and fluency of writing. They were not involved in research ideation, experimental design, data analysis, or interpretation. The authors take full responsibility for all content.

Table 15: Quantitative comparison with more motion-centric metrics.

| Method | Pose Confidence ↑ | Temporal Smoothness ↓ | Contact-Violation ↓ | Action Recognizability ↑ |
|---|---|---|---|---|
| VideoCrafter2 | 0.831 | 0.155 | 4.06% | 89.70% |
| LaVie | 0.865 | 0.138 | 3.25% | 92.14% |
| Mochi 1 | 0.886 | 0.121 | 2.04% | 96.75% |
| CogVideoX | 0.882 | 0.124 | 2.17% | 96.47% |
| HunyuanVideo | 0.894 | 0.119 | 1.89% | 97.28% |
| Wan2.1 | 0.893 | 0.117 | 1.62% | 97.56% |
| Ours | **0.911** | **0.103** | **1.08%** | **99.45%** |

Table 16: User study with more motion-aware metrics.

| Method | Motion Plausibility ↑ | Action Correctness ↑ |
|---|---|---|
| VideoCrafter2 | 0.85% | 2.04% |
| LaVie | 3.37% | 2.14% |
| Mochi 1 | 12.18% | 11.59% |
| CogVideoX | 12.36% | 14.82% |
| HunyuanVideo | 16.45% | 16.99% |
| Wan2.1 | 17.47% | 17.14% |
| Ours | **37.32%** | **35.28%** |

Table 17: Effect of the data from different views in our dataset.

| Model | FVD ↓ | CLIPSIM ↑ |
|---|---|---|
| w/o side-view data | 1172 | 0.2959 |
| w/o back-view data | 1165 | 0.2963 |
| Ours | **1096** | **0.3031** |

Table 18: Ablation study on the human animation task.

| Model | SSIM ↑ | PSNR ↑ | LPIPS ↓ | FVD ↓ | FID ↓ |
|---|---|---|---|---|---|
| w/o Struc. branch | 0.677 | 20.603 | 0.266 | 682.1 | 42.74 |
| w/o HADC | 0.682 | 20.716 | 0.247 | 627.9 | 39.55 |
| w/o Tracking loss | 0.688 | 20.785 | 0.218 | 594.1 | 36.64 |
| Ours | **0.691** | **20.802** | **0.209** | **568.3** | **34.76** |

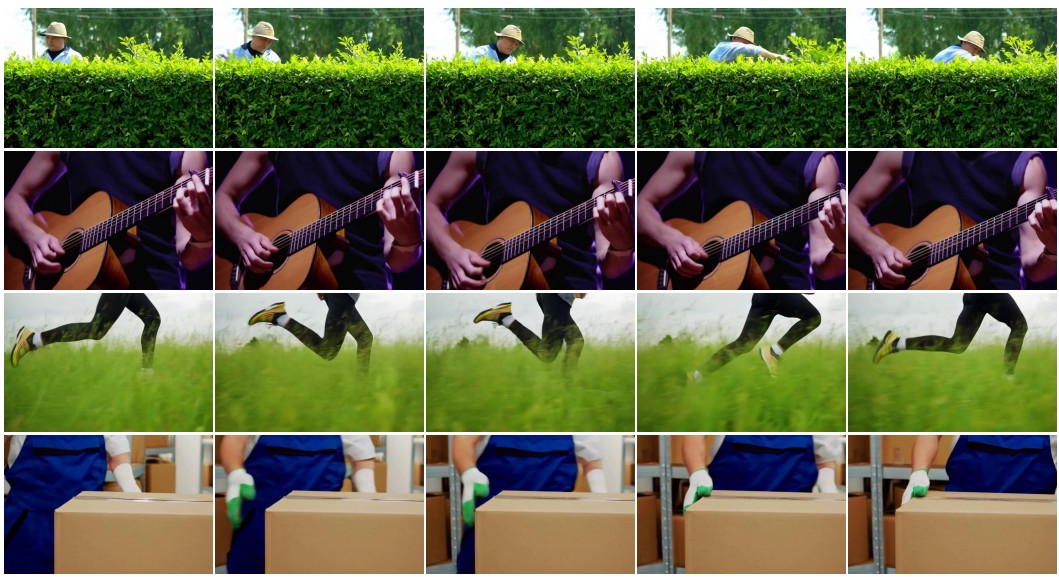

Figure 18: Visual results in which the human body is partially occluded.

