# OpenReview forum: "MoSA: Motion-Coherent Human Video Generation via Structure-Appearance Decoupling"
_ICLR.cc/2026/Conference — ICLR 2026 Poster_

### Official Review · Reviewer_4dkv · 2025-10-29

**Soundness:** 3
**Presentation:** 3
**Contribution:** 3
**Rating:** 6
**Confidence:** 3

**Summary:**

This paper proposed a motion-coherent human video generation model, called MoSA. Formally, MoSA decouples the structure and appearance generation through a 3D structure generation and an appearance generation branch. The structure generation branch comprises a text-to-motion Transformer, structure-gen blocks, and Human-Aware Dynamic Control (HADC) blocks, while the appearance generation is based on a pretrained video generation model. Mask loss and tracking loss are additionally incorporated to improve the performance. Detailed experiments show that the proposed method enjoys visually coherent results. Moreover, this paper proposed a new human video dataset, called MoVid, including 30K human motion videos exhibiting diverse action categories and complex motions.

**Strengths:**

1. This paper proposed an interesting idea for text-to-human generation, unifying both 3D text-to-motion and controllable video generation.
2. The writing quality of this paper is good, while qualitative and quantitative experiments are sufficient to cover SOTA video generation and user studies.
3. The proposed dataset, MoVid, seems to be useful to the community.

**Weaknesses:**

1. As detailed in Appendix.D, both the video generation model and text-to-motion transformer are frozen during the training. It would be helpful and interesting to investigate whether the text-to-motion model can also be enhanced with this end-to-end training strategy.

2. Although this paper included detailed experiments, some results are still lacking. For example, a qualitative ablation study for HADC and its visual outcomes can help the readers to understand the effect of HADC better.

3. For the comparison of human animation methods, the authors only compare to Animate Anyone, Champ, and HumanVid with inferior video backbones. More in-depth discussion and comparison are required, such as the Wan-based Uni3C (text-to-smpl can be used to simulate the Uni3C's input conditions as discussed in their paper).

**Questions:**

I recommend that the authors further compare their method to the SOTA human animation methods, such as Uni3C, to confirm their performance. Morevoer, an interesting discussion should be included for the unified training of both text-to-motion and controllable video generation.

---

> ### Author Response · Authors · 2025-11-19
>
> **Weakness 1**: Unified training of both text-to-motion and controllable video generation
>
> **Answer 1**: Thank you for your valuable suggestion! We believe that this unified training paradigm is highly worth exploring. Current text-to-motion model is pretrained on 3D motion datasets[1,2] that contain paired text descriptions and 3D human joint sequences. The unified training strategy would require a large-scale human motion dataset that provides aligned text–3D joint–video triplets. We surveyed existing open-source datasets and found that most of them contain only text–3D joint pairs or text–video pairs. Motion-X++ [3] provides video–3D joint pairs, but its videos are primarily indoor scenes with simple motions. Only a few closed-source datasets [4] may fully meet the requirements.
>
> Furthermore, adopting such a unified training paradigm based on our proposed MoVid dataset would require a substantial amount of time to annotate 3D human joints. We believe that as long as the dataset is sufficiently large and of high quality, such a unified training paradigm can improve model performance. However, if the training data are limited in scale or of low quality, the model may overfit to specific motion patterns, which could in turn lead to degraded performance.
>
> We strongly agree with this suggestion and will further validate the effectiveness of the unified training paradigm by expanding the 3D joint annotation on the proposed Movid dataset.
>
>
>
> **Weakness 2**: Qualitative ablation study for HADC
>
> **Answer 2**: Thank you for your comments! Due to the page limitations of the initial submission, the qualitative ablation study for HADC is provided in Fig. 13(a) of the appendix. When the HADC modules are removed, the generated human structure maintains a roughly correct body layout but loses the capacity for fine-grained motion control. Similarly, without the supervision of the motion loss function, $\mathcal{L}_m$, the details of the corresponding human motion are rendered to be unrealistic. In contrast, our full method successfully generates realistic yoga poses with plausible human structure. This clearly demonstrates that both the HADC modules and the $\mathcal{L}_m$ loss are indispensable for generating high-quality, physically plausible human motion.
>
> An example of visual outcomes is shown in the top-right portion of Fig. 2, demonstrating that introducing the HADC modules enhances the fine-grained control of human motion through the structural guidance. We additionally add a visualization of the HADC outputs in Fig. 14 in the paper's appendix, highlighting the human part in the final results.
>
>
>
> **Weakness 3**: Comparison with more work on human animation task
>
> **Answer 3:** Thank you for your suggestion! We conduct a quantitative comparison with recent human animation method Uni3C, as shown in the table below. Tab. 12 has also been updated in the paper, adding a comparison and reference to Uni3C. The results show that the proposed method also achieves strong performance on the human animation task. We also add these results to the paper's appendix.
>
> Table R2-1: Comparison with Uni3C on the human animation task.
>
> | Method | SSIM ↑    | PSNR ↑     | LPIPS ↓   | FVD ↓     | FID ↓     |
> | ------ | --------- | ---------- | --------- | --------- | --------- |
> | Uni3C  | 0.687     | 20.779     | 0.221     | **562.7** | 35.11     |
> | Ours   | **0.691** | **20.802** | **0.209** | 568.3     | **34.76** |
>
>
>
> [1] Go to Zero: Towards Zero-shot Motion Generation with Million-scale Data
>
> [2] Generating Diverse and Natural 3D Human Motions From Text
>
> [3] Motion-X++: A Large-Scale Multimodal 3D Whole-body Human Motion Dataset
>
> [4] LaserHuman: Language-guided Scene-aware Human Motion Generation in Free Environment

---

> ### Comment · Reviewer_4dkv · 2025-11-24
> **Thanks for the response**
>
> Thanks for the response from the authors. The rebuttal addressed most of my concerns. For Answer 1, I have an extended question about why 3D joint annotation is required. What about training the overall pipeline end-to-end (i.e., supervise the text-to-motion via video outcomes directly)?

---

> ### Author Response · Authors · 2025-11-24
> **Thanks for the valuable feedback and insightful question**
>
> Thank you once again for your valuable feedback and insightful question! We sincerely hope that our responses below address your concern.
>
> First, if we use 3D human structure as guidance and directly train the overall pipeline, due to the lack of 3D joint annotations, we can only use the ground truth video $V$ to supervise the generated video $V'$. However, our experiment has shown that for a given text prompt $p$, since the human structure $g_s$ generated by the 3D structure transformer is **not aligned** with the human structure in the ground truth video $V$, and there is a lack of 3D joint annotations corresponding to the ground truth video $V$ to **constrain $g_s$**, the training process exhibits poor convergence, and the model struggles to learn how to synthesize subsequent video appearances conditioned on the generated human structure $g_s$.
>
> Furthermore, if we use 2D human structure as guidance and only use the ground truth video $V$ to supervise the generated video $V'$, the problem of misalignment between the generated 2D human structure $g_s$ and the human structure in the ground truth video $V$ will still occur. Building on this, using the 2D human structure (*e.g.* skeleton) extracted from $V$ as supervision to constrain $g_s$ could alleviate this misalignment problem. However, as shown in Tab. 4 and Fig. 5 of our paper, we found that directly using the generated 2D structure disregards depth information in 3D space, making it difficult for the model to preserve spatial coherence in scenarios with limb occlusion. For example, as shown in the second row on the right of Fig. 5, the model incorrectly places the right leg behind the left, resulting in an implausible body structure.
>
> In conclusion, we believe that this training paradigm built upon our decoupling paradigm and MoVid dataset holds substantial potential for further exploration. Nonetheless, it requires additional labeled human-structure supervision in MoVid dataset to adequately constrain the intermediate generated structure. Ideally, incorporating 3D structural supervision (*e.g.* 3D joints) would better exploit spatial depth information, thereby maintaining spatial coherence in challenging scenarios involving limb occlusion.

---

> > ### Comment · Reviewer_4dkv · 2025-11-28
> > **I retain my score**
> >
> > Thanks for the response. It is a pity that the proposed system can not be learned end-to-end without 3D joint annotations, but the overall contributions remain solid. Thus, I retain my initial score of borderline accept.

---

> > > ### Author Response · Authors · 2025-11-28
> > >
> > > Thank you sincerely for your valuable time to review our paper and providing such insightful feedback. Obtaining accurate 3D joint annotations for the proposed MoVid dataset will require substantial additional time. With the continued progress of 3D reconstruction techniques, such as SAM3D Body, we believe that generating high-quality 3D annotations will become increasingly feasible, and we will further validate the effectiveness of the unified training paradigm by expanding the 3D joint annotation on the proposed Movid dataset, thus enriching the currently used 3D motion datasets.

---

### Official Review · Reviewer_gVyP · 2025-10-30

**Soundness:** 3
**Presentation:** 3
**Contribution:** 2
**Rating:** 6
**Confidence:** 4

**Summary:**

MoSA proposes a structure–appearance decoupling pipeline for human video generation: starting from a motion-specific prompt, a pretrained 3D structure transformer predicts 3D keypoint sequences that are projected to 2D skeletons and injected via specialized structure blocks to guide a DiT-based appearance model. To turn sparse skeleton cues into precise motion control, the authors add Human-Aware Dynamic Control (HADC) with a learned mask loss that assigns spatially varying weights over human regions. Motion realism is further enforced with a dense tracking loss for temporal coherence and a 3D human–scene contact constraint that discourages interpenetration. They also curate MoVid, a large-scale human video dataset emphasizing diverse and complex motions. Empirically, MoSA plugs into strong base models (e.g., CogVideoX, Wan 2.1) and shows consistent gains over prior work, and the paper additionally provides an image-to-video (I2V) variant in the appendix.

**Strengths:**

1. This paper proposes to use 3D to 2D structure guidance, and HADC integrates cleanly into DiT backbones. Experiments shows that it works with CogVideoX and Wan 2.1. Removing the structure branch or replacing 3D with direct 2D pose generation degrades plausibility (missing/occluded limbs).

2. To enhance motion coherence and physics, dense-tracking and contact terms are proposed and proved to be effect.

3. Dataset contribution addresses a real gap (full-body, complex motion) and is used in comparisons.

**Weaknesses:**

1. Is projected 2D structure necessary for T2V? The authors argue that generating 3D keypoints and projecting to 2D improves plausibility and occlusion handling over directly predicting 2D skeletons. Ablations show failures like missing limbs when using a 2D structure generator, which supports that claim. However, the current metrics (FVD, CLIP similarity, and some VBench dimensions) are not targeted at structural correctness, so they don’t decisively isolate the benefit of 3D to 2D conditioning for T2V. Please complement FVD/CLIP/VBench with motion-centric metrics, e.g., pose detector, self-occlusion correctness, temporal smoothness/jerk on tracked keypoints, foot-sliding/contact-violation rate, and action recognizability via a pretrained classifier, to more directly validate structure quality. FVD, CLIP similarity and VBench-style scores are useful but still loosely connected to human motion fidelity, so they may not fully capture improvements in structure plausibility or contact realism. I recommend adding a small targeted user study (preference on “motion plausibility” and “action correctness”) and reporting standardized, motion-aware metrics as noted above to strengthen claims beyond general perceptual quality.

2. Training–inference structure gap. During training, the 3D structure transformer is excluded and the model conditions on 2D skeletons extracted from the ground-truth video, whereas at inference it conditions on 2D projections of generated 3D structures. This creates a distribution shift that the paper doesn’t directly address. While Table 13 suggests robustness to user-specified fixed or time-varying cameras at inference, it would help to report results when training includes similar view randomization.

3. Representation limits (sparse skeletons & multi-person). The approach is bottlenecked by the sparsity of skeletons: this paper explicitly notes that skeleton guidance lacks expressiveness for fine-grained control, requiring HADC to propagate structure cues into denser motion regions. This leaves open limitations for hands, garments, and challenging multi-person occlusions. The paper shows half-body and multi-person qualitative results, but a more systematic analysis of failure modes (e.g., hand interactions, inter-person contact) would clarify scope. A brief discussion of these limits and whether denser cues (SMPL-X joints, hand keypoints) would slot into the same framework would help readers understand when MoSA may still struggle.

4. The authors do provide an image-to-video (I2V) variant in Appendix H to report an I2V “human animation” setting where the 3D structure transformer is removed. Is the good performance from better pretrained model (CogVideoX and Wan 2.1) or the model design? Because most of the compared methods use a U-net based pretraining model.

**Questions:**

Please see the weaknesses. Please correct me if my understanding about some points are wrong.

Overall, it is a good attempt to incorporate human skeleton and motion generation methods into T2V human video generation. However, I still have some concerns about the evaluation part.

---

> ### Author Response · Authors · 2025-11-19
>
> **Weakness 1**: Evaluation on more metrics
>
> **Answer 1**: Thank you for your valuable suggestions! We design more motion-centric metrics and add these evaluation metrics to the ablation study of the structure-appearance decoupling paradigm, as shown in Table R1-1. Furthermore, we use these metrics to compare existing T2V models with our approach, and the corresponding results are presented in Table R1-2. The Pose Confidence metric is computed by applying the DWPose [1] model to extract human keypoints from the generated videos and averaging their confidence scores. Higher confidence indicates more accurate pose estimation, which further shows the plausibility of the generated human poses, avoiding structural failures such as missing limbs. Temporal Smoothness is derived by calculating the mean jerk of tracking points within human regions, where lower values correspond to smoother motion. The Contact-Violation metric is evaluated using Qwen3-VL, a powerful multimodal model equipped with extensive world knowledge. It identifies physically implausible artifacts such as interpenetrations in the generated videos and computes the corresponding violation rate. Action Recognizability is also assessed using Qwen3-VL, which determines whether the generated actions faithfully match those specified in the text prompt, such as the text prompt “walks to the blue trampoline and jumps” in Fig. 3. Motion Plausibility and Action Correctness are the evaluation criteria used in the user study.
>
> Moreover, we have conducted a user study, as reported in Tab. 9 in the paper's appendix, to assess the motion quality of the generated videos. To more comprehensively demonstrate the advantages of our approach, we further evaluate the motion plausibility and action correctness of the generated videos, with the corresponding results presented in Table R1-3. These motion-centric metrics provide additional evidence that our method exhibits clear superiority in generating high-quality human motion. We also add these results to the paper's appendix.
>
>
>
> Table R1-1: Effect of the decoupling paradigm, evaluating on more motion-centric metrics.
>
> | Structure Generation Branch | Pose Confidence ↑ | Temporal Smoothness ↓ | Contact-Violation ↓ | Action Recognizability ↑ | Motion Plausibility ↑ | Action Correctness ↑ |
> | --------------------------- | ----------------- | --------------------- | ------------------- | ------------------------ | --------------------- | -------------------- |
> | ×                           | 0.885             | 0.122                 | 2.04%               | 96.85%                   | 5.09%                 | 8.24%                |
> | 2D Structure Generation     | 0.896             | 0.114                 | 1.60%               | 98.42%                   | 15.13%                | 15.60%               |
> | Ours                        | **0.911**         | **0.103**             | **1.08%**           | **99.45%**               | **79.78%**            | **76.16%**           |
>
>
>
> Table R1-2: Quantitative comparison with more motion-centric metrics.
>
> | Method        | Pose Confidence ↑ | Temporal Smoothness ↓ | Contact-Violation ↓ | Action Recognizability ↑ |
> | ------------- | ----------------- | --------------------- | ------------------- | ------------------------ |
> | VideoCrafter2 | 0.831             | 0.155                 | 4.06%               | 89.70%                   |
> | LaVie         | 0.865             | 0.138                 | 3.25%               | 92.14%                   |
> | Mochi 1       | 0.886             | 0.121                 | 2.04%               | 96.75%                   |
> | CogVideoX     | 0.882             | 0.124                 | 2.17%               | 96.47%                   |
> | HunyuanVideo  | 0.894             | 0.119                 | 1.89%               | 97.28%                   |
> | Wan2.1        | 0.893             | 0.117                 | 1.62%               | 97.56%                   |
> | Ours          | **0.911**         | **0.103**             | **1.08%**           | **99.45%**               |
>
> Table R1-3: User study with more motion-aware metrics.
>
> | Method        | Motion Plausibility ↑ | Action Correctness ↑ |
> | ------------- | --------------------- | -------------------- |
> | VideoCrafter2 | 0.85%                 | 2.04%                |
> | LaVie         | 3.37%                 | 2.14%                |
> | Mochi 1       | 12.18%                | 11.59%               |
> | CogVideoX     | 12.36%                | 14.82%               |
> | HunyuanVideo  | 16.45%                | 16.99%               |
> | Wan2.1        | 17.47%                | 17.14%               |
> | Ours          | **37.32%**            | **35.28%**           |

---

> ### Author Response · Authors · 2025-11-19
>
> **Weakness 2**: Structure guidance during training and inference
>
> **Answer 2**: Thank you for your comments! We kindly remind that during training, the skeleton sequence extracted from the ground-truth video is directly used as the structural guidance, as stated in line 319 of the paper. Therefore, the training process does not involve selecting a camera view or performing any projection step. Furthermore, the 2D skeletons extracted during training and the skeletons rendered during inference share the exact same representation, as shown in the skeleton visualization in the “Ours” row of Fig. 5. This allows the model to seamlessly leverage the structural guidance generated by the 3D structure transformer at inference time, thereby improving its ability to generate complex motions.
>
> Furthermore, as illustrated in Fig. 7 of the appendix, the proposed MoVid dataset contains human videos captured from multiple viewpoints, effectively providing a form of “view diversity” during training. This helps the trained model produce high-quality results across different camera views. To further validate the contribution of the MoVid dataset in this regard, we conducted experiments where side-view and back-view videos in our dataset are removed during training. The quantitative results, shown in Table R1-4, support the above statements. We also add these results to the paper's appendix.
>
> Table R1-4: Effect of the data from different views in our dataset.
>
> | Model              | FVD ↓    | CLIPSIM ↑  |
> | ------------------ | -------- | ---------- |
> | w/o side-view data | 1172     | 0.2959     |
> | w/o back-view data | 1165     | 0.2963     |
> | Ours               | **1096** | **0.3031** |
>
>
>
> **Weakness 3**: More discussions on human representations
>
> **Answer 3**: Thanks for the insightful suggestion! Generating complex inter-person contact and fine-grained hand interactions has long been a challenging problem in video generation. Built upon a structure–appearance disentanglement paradigm，our method substantially improves the quality of multi-person interactions and fine-grained motion synthesis. As shown in Fig. 18, the girl’s bowstring-pulling action in the second row and the two-person jumping and ball-contesting scene in the third row demonstrate clear enhancements.
>
> Nevertheless, we found that generating highly intricate hand motions remains a challenge, where current methods may still produce artifacts such as distorted or blurred finger structures. Note that, our structure-appearance disentanglement paradigm is compatible with incorporating denser structural cues, such as hand keypoints. This inherent compatibility offers a promising pathway to enhance the realism of finger-level actions by integrating such detailed guidance.
>
> Our further investigation reveals that the primary obstacle lies in the training data. Existing motion datasets used for training 3D structure transformers typically include only SMPL body joints. Consequently, incorporating hand keypoints would necessitate augmenting these datasets with additional 3D annotations for hand joints to enable effective model training. We believe this is a worthwhile direction to explore and will pursue it in our future work, with extending the proposed MoVid dataset.

---

> ### Author Response · Authors · 2025-11-19
>
> **Weakness 4**: More comparisons on the human animation setting
>
> **Answer 4**: Thank you for your suggestions! A stronger video foundation model offers a higher performance baseline. The strong performance of our approach stems not only from the more capable pretrained model but also from our carefully designed architecture. To substantiate this claim, we conducted a series of ablation studies, and the results are presented in Table R1-5. Specifically, the first row corresponds to removing the structural branch, where the pose latents encoded by the VAE are added to the video latents. The second and third rows represent ablations of the proposed HADC modules and the dense tracking constraint, respectively. Since the current human animation setting involves minimal human–environment interaction, the proposed 3D contact constraint is not included in the ablation analysis. The results demonstrate that, beyond the stronger video foundation model, our carefully designed architecture further improves overall performance.
>
> Furthermore, for a fairer comparison, we also conduct a quantitative comparison with the recent human animation method Uni3C [2] in Table R1-6, which is based on the Wan 2.1 pre-trained model and uses the same DiT architecture. The results further demonstrate the effectiveness of the proposed method. We also add these results to the paper's appendix.
>
> Table R1-5: Ablation study on the human animation task.
>
> | Model             | SSIM ↑    | PSNR ↑     | LPIPS ↓   | FVD ↓     | FID ↓     |
> | ----------------- | --------- | ---------- | --------- | --------- | --------- |
> | w/o Struc. branch | 0.677     | 20.603     | 0.266     | 682.1     | 42.74     |
> | w/o HADC          | 0.682     | 20.716     | 0.247     | 627.9     | 39.55     |
> | w/o Tracking loss | 0.688     | 20.785     | 0.218     | 594.1     | 36.64     |
> | Ours              | **0.691** | **20.802** | **0.209** | **568.3** | **34.76** |
>
> Table R1-6: Comparison with Uni3C on the human animation task.
>
> | Method | SSIM ↑    | PSNR ↑     | LPIPS ↓   | FVD ↓     | FID ↓     |
> | ------ | --------- | ---------- | --------- | --------- | --------- |
> | Uni3C  | 0.687     | 20.779     | 0.221     | **562.7** | 35.11     |
> | Ours   | **0.691** | **20.802** | **0.209** | 568.3     | **34.76** |
>
>
>
> [1] Effective Whole-body Pose Estimation with Two-stages Distillation
>
> [2] Uni3C: Unifying Precisely 3D-Enhanced Camera and Human Motion Controls for Video Generation

---

### Official Review · Reviewer_74M1 · 2025-11-01

**Soundness:** 2
**Presentation:** 3
**Contribution:** 3
**Rating:** 4
**Confidence:** 5

**Summary:**

This paper presents a novel framework which decouples the process of human video generation into structure generation and appearance generation.

**Strengths:**

The motivation is interesting, and decoupling structure from appearance addresses a long-standing challenge in text-to-human video generation.

Experiments on multiple benchmarks show substantial improvements compared with existing methods.

**Weaknesses:**

Does the proposed method ensure physical realism—for example, the deformation of the trampoline shown in Figure 3?

The overall framework just combines known ideas (DiT backbone + structural priors + control modules), thus the innovation is mainly architectural integration.

This paper adopts a mask-based generation approach. How does this method handle problem where the human body is partially occluded?

**Questions:**

please refer to weakness

---

> ### Author Response · Authors · 2025-11-19
>
> **Weakness 1**: Whether the proposed method ensures physical realism
>
> **Answer 1**: Thank you for your comment! In terms of physical realism, our method demonstrates strong performance on two key fronts:  human body structure and surrounding environment.
>
> First, for the physical realism of human body structure, our proposed structure-appearance disentangled generation paradigm plays a crucial role. This paradigm ensures the generation of physically plausible and complete human structures, meanwhile preventing severe artifacts such as missing legs shown in Fig. 1. By guaranteeing structural integrity and plausibility, our model is further empowered to generate more complex and challenging human motions, and we demonstrate its effectiveness through extensive quantitative and qualitative comparisons.
>
> Second, for the physical realism of the surrounding environment, our method effectively preserves the strong generative capabilities of the underlying video models, CogVideoX and Wan2.1. This ensures that the physical consistency is maintained during generation. For example, in the supplementary materials (video.mp4), the black ball at approximately the 52-second mark maintains its physical rigidity while in motion, exhibiting no unrealistic distortions. We have also added the corresponding visual results for this example in Fig. 10 in the appendix for reference, highlighted in blue. Similarly, the subtle deformation of the trampoline under force is realistically rendered around the 25-second mark.
>
>
>
> **Weakness 2**: The innovation of our work
>
> **Answer 2**: Thanks for this comment! Our main contribution lies in the structure–appearance decoupling generation paradigm, which effectively mitigates human generation issues in advanced text-to-video models (e.g., HunyuanVideo and Wan 2.1), such as human-body structural distortion and unrealistic motion. To the best of our knowledge, our work is an original effort on this issue, and the paradigm is novel for human video generation.
>
> For methodologies, the proposed framework is not simply architectural integration. We introduce Human-Aware Dynamic Control modules to enhance the fine-grained controllability of structural guidance for human motion. We further propose a 3D contact constraint and a dense tracking loss to improve generation quality. Table 3, 5, and 6 in our paper demonstrate the effectiveness of these proposed methods.
>
>
>
> **Weakness 3**: How to handle occlusion situations
>
> **Answer 3**: Thanks for the insightful comment! Our proposed method is also compatible with scenarios where the human body is partially occluded.
>
> First, we kindly note that the human mask is used exclusively as an auxiliary supervision during the training phase to train our proposed Human-Aware Dynamic Control (HADC) module, and it is not used during inference.
>
> Specifically, the core role of the HADC modules is to enhance the model's fine-grained control over human motion. During training, we introduce a mask loss $L_m$ calculated from the ground-truth human masks. This process guides the model to learn how to effectively propagate the sparse structural guidance $s$ to the corresponding **visible regions** of the human body in the image. Through this supervision, the model learns the association between the skeletal structure and the actual visible body parts.
>
> Consequently, this mechanism could handle scenarios with partial occlusions. During training, if a part of the human body is occluded by an object (e.g., a leg hidden behind a table), the corresponding ground-truth human mask for that area will be absent. The model learns that even when the structural guidance $s$ indicates that a leg should be present, it should avoid generating the leg’s appearance in regions where it is actually occluded. In other words, the HADC module learns to render the visual appearance only in the **non-occluded areas** corresponding to the structural guidance. This capability is demonstrated on the left side of Fig. 5 in our paper, where our model properly handles a person being occluded by objects.
>
> To further substantiate the effectiveness of our approach, we add additional visual results in Fig. 19 in the appendix. These results showcase more diverse occlusion scenarios and clearly demonstrate that MoSA can generate high-quality and physically plausible videos even when partial occlusions are present. We also add these discussions to the paper's appendix.

---

> ### Author Response · Authors · 2025-11-28
>
> Dear Reviewer 74M1:
>
> Thank you very much for your valuable time in reviewing our manuscript.
>
> As the discussion period is approaching its end, we would like to kindly check if you have had the opportunity to review our detailed responses to your comments. We would greatly appreciate it if you could let us know whether our clarifications adequately address your concerns regarding the paper’s innovation and methodological details. If there are any remaining questions, we would be sincerely grateful for the chance to provide further clarification and will respond with our utmost effort.
>
> Thank you again for your time and consideration.
>
> Best regards,
>
> Authors of Paper 9376

---

### Official Review · Reviewer_sL3c · 2025-11-01

**Soundness:** 3
**Presentation:** 3
**Contribution:** 3
**Rating:** 6
**Confidence:** 4

**Summary:**

This paper introduces a video generation method with finegrained human motion controlled from text prompt. The key idea is to guide the appearance generation on a set of 2D keypoints, projected from 3d keypoints, with the assistant of a spatial variation control weight regions. It also introduces a large scale dataset with 30k diverse and complex human motion. The proposed method shows superior results over prior baselines.

**Strengths:**

1. Decoupling of motion and appearance to guide the video generation to more complex human motion makes sense
2. The Movid dataset is an important contribution to the field

**Weaknesses:**

1. The method uses motion corerspondences to constrain video consistency. These correspondences can be noisy, especially for textureless regions. Since these correspondences are only estimated from RGB frames, ie., the final output within a diffusion process, unrolling the gradient from multi-step diffusion is expensive.
2. It is unclear how the 3D contract constraint is differentiable.

**Questions:**

Please respond to the weekness section

---

> ### Author Response · Authors · 2025-11-19
>
> **Weakness 1**：Accuracy and cost of motion correspondence
>
> **Answer 1**: Thanks for the comments! The motion correspondences are extracted using the widely adopted tracking model CoTracker3 [1], which already demonstrates strong performance. For instance, its occlusion accuracy exceeds 90% on multiple benchmarks. Furthermore, before utilizing the motion correspondences, we filter out all tracking points with low confidence. Although a small amount of noise may still remain, the extracted correspondences effectively capture the underlying human motion patterns, which in turn contributes positively to the model’s performance. This effect is further supported by the ablation results presented in Tab. 6.
>
> Moreover, we kindly note that, during training, computing this constraint does not require running the full multi-step diffusion process. Specifically, during training, a random noise $\epsilon \sim \mathcal{N}(0, I)$ is sampled and added to the ground truth video latents $z_0$, i.e.,
>
> $$z_t = \sqrt{\bar{\alpha}_t} z_0 + \sqrt{1-\bar{\alpha}_t} \epsilon, ~ \epsilon \sim \mathcal{N}(0, I) ,$$
>
> where $\bar{\alpha}_t$ represents a predefined noise schedule. Next, the generated video latents $\hat{z}_0$ can be obtained by subtracting the model-predicted noise $\hat{\epsilon}$ from the noised latent $z_t$, i.e.,
>
> $$\hat{z}_0  = \frac{z_t - \sqrt{1 - \bar{\alpha}_t} \hat{\epsilon} }{\sqrt{\bar{\alpha}_t}} .$$
>
> Then we can decode $\hat{z}_0$ with VAE to get the generated video $V'$, and extract the tracking correspondence between the ground truth video $V$ and the generated video $V'$, and calculate the dense tracking loss. Therefore, this procedure does not require executing the full multi-step diffusion process.

---

> ### Author Response · Authors · 2025-11-19
>
> **Weakness 2**: 3D contract constraint differential process
>
> **Answer 2**: Thank you for your valuable suggestion! We explain the differentiation process of 3D contract constraint in detail below. We've also included this process in the appendix, highlighted in blue.
>
> For the f-th frame, the 3D contact loss is defined as follows, as shown in Eq. 10 of the paper:
>
> $$
> \mathcal{L}_{cont}^{f} =
> \begin{cases}
> \\sum\\limits\_\{h \\in H\_{f}\^{-} \} | SDF(h) - \tau |,  \& \\text\{if \} \|H\_\{f\}\^\{-\}\| > 0 \\\\
> \\min\_\{h \\in H\_\{f\}\} SDF(h) - \\tau, & \text{otherwise}.
> \end{cases}
> $$
>
>
> The propagation path is as follows, where $h$ denotes a human point, $V'$ is the generated video, $\theta$ denotes the model parameters.
>
> $$
> \mathcal{L}_{cont}^{f}  → SDF(h) → h → V' → θ
> $$
>
> (1) The process begins with the loss function $\mathcal{L}_{cont}^{f}$ itself. Since $\tau$ is set to 0, for a human point $h$, the loss computation can be simplified to:
>
> $$
> \mathcal{L}_{cont}^{\text{f,h}} = max(0, - SDF(h))
> $$
>
> (2) The partial derivative of this loss with respect to the SDF value is non-zero only in the case of interpenetration:
>
> $$
> \frac{\partial \mathcal{L}_{cont}^{\text{f,h}}}{\partial SDF(h)} =
> \begin{cases}
> -1, & \text{if } SDF(h) < 0, \\\\
> 0, & \text{otherwise}.
> \end{cases}
> $$
>
> (3) The next step is to propagate the gradient from the SDF value to the 3D coordinates $(x, y, z)$ of the point $h$. The gradient of a scalar field like the SDF with respect to a point's coordinates is a vector:
>
> $$
> \nabla_h SDF = \frac{\partial SDF(h)}{\partial h}
> $$
>
> This gradient vector provides the most efficient direction to displace the point $h$ in order to move it towards the exterior of the scene mesh, thereby resolving the penetration.
>
> Using the chain rule, we combine the gradients from the previous two steps to compute the gradient of the loss with respect to the 3D coordinates of the human point $h$:
>
> $$
> \\frac\{\\partial \\mathcal\{L\}\_\{cont\}\^\{\\text\{f,h\}\}\}\{\\partial h\} = \\frac\{\\partial \\mathcal\{L\}\_\{cont\}\^\{\\text\{f,h\}\}\}\{\\partial SDF(h)\} \\cdot \\frac\{\\partial SDF(h)\}\{\\partial h\}
> $$
>
> (4) Then, the gradient $\frac{\partial \mathcal{L}_{cont}^{\text{f,h}}}{\partial h}$ can be backpropagated through the layers of VGGT [2] to obtain the gradient with respect to its input, i.e. the generated video frame $V'$:
>
> $$
> \\frac\{\\partial \\mathcal\{L\}\_\{cont\}\^\{\\text\{f,h\}\}\}\{\\partial V'\} = \\frac\{\\partial \\mathcal\{L\}\_\{cont\}\^\{\\text\{f,h\}\}\}\{\\partial h\} \\cdot \\frac\{\\partial h\}\{\\partial V'\}
> $$
>
> (5) Finally, the generated video frame $V'$ is the output of our model, $V' = MoSA(inputs; θ)$. With the gradient $\frac{\partial \mathcal{L}_{cont}^{\text{f,h}}}{\partial V'}$ computed, we can continue the backpropagation through the MoSA network to obtain the gradients with respect to all of its learnable parameters $θ$:
>
> $$
> \\frac\{\\partial \\mathcal\{L\}\_\{cont\}\^\{\\text\{f,h\}\}\}\{\\partial \\theta\} = \\frac\{\\partial \\mathcal\{L\}\_\{cont\}\^\{\\text\{f,h\}\}\}\{\\partial V'\} \\cdot \\frac\{\\partial V'\}\{\\partial \\theta\}
> $$
>
> These final gradients are then used by the optimizer to update the model's parameters.
>
> [1] Cotracker3: Simpler and Better Point Tracking by Pseudo-Labelling Real Videos
>
> [2] VGGT: Visual Geometry Grounded Transformer

---

### Author Response · Authors · 2025-12-03
**Rebuttal Summary**

**Dear Program Chairs, Senior Area Chairs, Area Chairs, and Reviewers,**

We sincerely appreciate your feedback and insights. We have carefully considered each concern and incorporated substantial updates. As the discussion phase concludes, we provide a concise summary of the responses and revisions made during the rebuttal.

**Reviewer sL3c (Rating: 6)**

**1**. Accuracy and Cost of Motion Correspondence

We clarified that motion correspondence quality is promoted using CoTracker3 with confidence filtering, as supported by the ablation in Table 6. We also described our efficient training strategy, which computes dense tracking loss via latent reconstruction to avoid multi-step diffusion overhead.

**2**. Differentiation of the Contact Constraint

We expanded the Appendix to include the full derivation of the 3D contact constraint differentiation, now provided in **Sec. B**.

**Reviewer 74M1 (Rating: 4)**

**1**. Preservation of Physical Realism

In response to this concern, we explained that our structure–appearance decoupling paradigm guarantees the integrity of human structural cues, while its integration with a strong video backbone preserves environmental consistency. These capabilities are further demonstrated through the newly added **Fig. 10**, which showcases physically plausible rigid object motion.

**2**. Handling of Occlusions

In response to the concern regarding occlusion handling, we clarified that our mask-supervised training strategy enables the Human-Aware Dynamic Control (HADC) module to render structural guidance only in visible regions, effectively managing partial occlusions without requiring masks during inference. To further substantiate this capability, we have included additional visual results representing diverse occlusion scenarios in **Fig. 19** of the Appendix.

**3**. Innovation of Our Work

In response to this concern, we clarified that our contribution centers on a novel structure–appearance decoupling paradigm designed to alleviate structural distortions in advanced video models. We further highlighted that our method extends beyond straightforward architectural integration by incorporating Human-Aware Dynamic Control modules and physics-based constraints. The effectiveness of these components is substantiated by the results reported in Tab. 3, Tab. 5, and Tab. 6.

**Reviewer gVyP (Rating: 6)**

**1**. Evaluations on More Metrics

We added motion-centric metrics, including Pose Confidence, Temporal Smoothness, and VLM-based assessments, with quantitative comparisons and user study results in **Tab. 14, Tab. 15, and Tab. 16** of the Appendix.

**2**. Details of Structure Guidance

We clarified that ground-truth skeleton sequences are used during training to maintain consistency with inference, and highlighted how the multi-view MoVid dataset improves view generalization. Corresponding quantitative results are included in **Tab. 17** of the Appendix.

**3**. Discussions on Human Representations

We showed that our method enhances multi-person and fine-grained interactions (Figure 18) and noted that current limitations in hand generation are due to scarce 3D hand annotations rather than architectural constraints. We also outlined plans to extend MoVid with hand keypoints for denser structural cues.

**4**. More Experiments on Human Animation Setting

A new ablation study in **Tab. 18** demonstrates that performance gains come from our architectural designs, not just the foundation model. We also added a comparison with Uni3C using the same DiT backbone in **Tab. 12**, with all results included in the Appendix.

**Reviewer 4dkv (Rating: 6)**

**1**. Unified Training Paradigm Attempt

We acknowledged the potential of a unified training paradigm but noted that the limited availability of aligned text–3D joint–video datasets hinders immediate implementation. We also discussed overfitting risks and plan to validate this paradigm by expanding 3D joint annotations in MoVid.

**2**. Qualitative Ablation Study for HADC

Existing **Fig. 13(a)** in the Appendix shows that the HADC module is crucial for fine-grained motion control. We added visualizations of HADC outputs in **Fig. 14** to explicitly illustrate its effects.

**3**. Comparison with Uni3C on Human Animation Task

We performed a comparison with Uni3C and updated **Tab. 12** in the Appendix, demonstrating that our method achieves strong performance on the human animation task.

We have incorporated all updates into the manuscript. Our work received consistent support from Reviewers 4dkv, gVyP, and sL3c for the contribution and novelty of our methodology and dataset. For Reviewer 74M1’s concerns on method details and innovation, we provided comprehensive clarifications, but no further feedback was received. We also had constructive discussions with Reviewer 4dkv, who acknowledged the effectiveness of our rebuttal.

Thank you again for your valuable time and consideration.

Sincerely,

**The Authors of Submission 9376**

---

### Meta-Review · Area_Chair_RT8S · 2026-01-05

**Summary:**

This paper receives 1 negative rating (4), and 3 positive ratings (6). Main concerns lie in 1) missing details (e.g., how to compute motion correspondences, how the 3D contract constraint is differentiable); 2) ablation studies to analyze individual modules (e.g. HADC); 3)limited generality (e.g. to sparse skeletons & multi-person, partially occluded skeletons); 4) weak baselines.

In authors' responses, all these concerns have been properly addressed with corresponding experiment results. AC found no additional concerns. Thus the decision is accept.

**Reviewer Concerns:**

Main concerns are: 1) missing details (e.g., how to compute motion correspondences, how the 3D contract constraint is differentiable); 2) ablation studies to analyze individual modules (e.g. HADC); 3)limited generality (e.g. to sparse skeletons & multi-person, partially occluded skeletons); 4) weak baselines.

All these concerns are properly addressed in the rebuttal in my opinion.

**Reviewer Scores:**

Reviewer 74M1 would increase his/her score since his/her concerns are well addressed in authors' rebuttal.

---

### Decision · Program_Chairs · 2026-01-26

Accept (Poster)